# Excitonic Mott insulator in a Bose-Fermi-Hubbard system of moiré WS$_2$/WSe$_2$ heterobilayer

Beini Gao [1,8], Daniel G. Suárez-Forero [1,8] ✉, Supratik Sarkar [1,8], Tsung-Sheng Huang[1], Deric Session[1], Mahmoud Jalali Mehrabad [1], Ruihao Ni [2], Ming Xie [3], Pranshoo Upadhyay[1], Jonathan Vannucci [1], Sunil Mittal[1], Kenji Watanabe [4], Takashi Taniguchi [4], Atac Imamoglu [5], You Zhou [2,6] & Mohammad Hafezi [1,7] ✉

Understanding the Hubbard model is crucial for investigating various quantum many-body states and its fermionic and bosonic versions have been largely realized separately. Recently, transition metal dichalcogenides heterobilayers have emerged as a promising platform for simulating the rich physics of the Hubbard model. In this work, we explore the interplay between fermionic and bosonic populations, using a WS$_2$/WSe$_2$ heterobilayer device that hosts this hybrid particle density. We independently tune the fermionic and bosonic populations by electronic doping and optical injection of electron-hole pairs, respectively. This enables us to form strongly interacting excitons that are manifested in a large energy gap in the photoluminescence spectrum. The incompressibility of excitons is further corroborated by observing a suppression of exciton diffusion with increasing pump intensity, as opposed to the expected behavior of a weakly interacting gas of bosons, suggesting the formation of a bosonic Mott insulator. We explain our observations using a two-band model including phase space filling. Our system provides a controllable approach to the exploration of quantum many-body effects in the generalized Bose-Fermi-Hubbard model.

The rich physics of the Hubbard model has brought fundamental insights to the study of many-body quantum physics[1]. Initially proposed for electrons on a lattice, different fermionic and bosonic versions of this model have been simulated in various platforms, ranging from ultracold atoms[2] to superconducting circuits[3]. Recently, bilayer transition metal dichalcogenides (TMDs) have become a versatile platform to study the Hubbard model thanks to the coexistence of several intriguing features such as the reduction of electron hopping due to the formation of moiré

lattice with large lattice constant, and the existence of both intra- and interlayer excitons. These characteristics have enabled the realization of numerous effects of many-body physics such as metal-to-Mott insulator transition[4–9], generalized Wigner crystals[10–14], exciton–polaritons with moiré-induced nonlinearity[15], stripe phases[16], light-induced ferromagnetism[17]. Moreover, there have been recent exciting perspectives of exploring such effects in light–matter correlated systems[3,18,19]. While typically the fermionic and bosonic versions of the Hubbard

[1]Joint Quantum Institute (JQI), University of Maryland, College Park, MD, USA. [2]Department of Materials Science and Engineering, University of Maryland, College Park, MD, USA. [3]Condensed Matter Theory Center, University of Maryland, College Park, MD, USA. [4]National Institute for Materials Science, Tsukuba, Japan. [5]Institute for Quantum Electronics, ETH Zurich, Zurich, Switzerland. [6]Maryland Quantum Materials Center, College Park, MD, USA. [7]Institute for Theoretical Physics, ETH Zurich, Zurich, Switzerland. [8]These authors contributed equally: Beini Gao, Daniel G. Suárez-Forero, Supratik Sarkar. ✉e-mail: dsuarezf@umd.edu; hafezi@umd.edu

model are explored independently, combining these two models in a single system holds intriguing possibilities for studying mixed bosonic-fermionic correlated states[20,21].

In this work, we demonstrate Bose–Fermi–Hubbard physics in a TMD heterobilayer. We independently control the population of fermionic (electronic) particles by doping with a gate voltage ($V_g$), and the population of bosonic (excitonic) states by pumping with a pulsed optical drive of intensity $I$. Harnessing these two control methods, we realize strongly interacting excitons. In particular, we show the incompressibility of excitonic states near integer filling by observing an energy gap in photoluminescence, accompanied by an intensity saturation. Remarkably, we observe the suppression of diffusion, a strong indication of the formation of a bosonic Mott insulator of excitons.

## Results
### Physical system and experimental design
To demonstrate these effects, we use a moiré lattice created by stacking two monolayers of $WS_2$ and $WSe_2$, with symmetric top and bottom gates. Figure 1a shows a schematic illustration of the heterobilayer device (see Supplementary Note 1 for details). Due to the type-II band alignment of the heterostructure (Fig. 1b), negative doping results in a population of electrons in the $WS_2$ subject to the moiré potential of the bilayer. The ratio between the density of this population and the density of moiré sites in the structure determines the so-called electronic filling factor ($\nu_e$). The optical pump results in the formation of an energetically favorable interlayer exciton (X)[22], by pairing between an electron in $WS_2$ and a hole in $WSe_2$ (represented in Fig. 1b). In order to explore different regimes of Bose–Fermi–Hubbard model, we control the bosonic and fermionic populations by changing $I$ and $V_g$, respectively. This can be compared to the ultracold atom implementation of Bose–Fermi mixture where the respective populations are fixed in each experiment[23]. Before discussing our experimental observation, we discuss three limiting cases that determine the phase space of our system, as indicated in Fig. 1c. The corresponding physical scenarios are represented in panels d–f. First, in the weak excitation limit and low electronic filling factor ($\nu_e$ ~0) regime,

the system's photoluminescence (PL) emission originates exclusively from the few X states in the quasi-empty lattice (panel d). This emission comes from excitons in lattice sites where they are the only occupant particles, namely, "single occupancy states" ($X_1$). Upon increasing $\nu_e$, the number of singly occupied sites decreases, and in the limiting case of $\nu_e \geq 1$, as represented in panel e, the optically generated excitons can only form in lattice sites already occupied by charged particles. In this case, the required energy to form the exciton increases due to the on-site Coulomb repulsion, and hence the PL emission has new branch with higher energy than the previous regime. Consequently, the PL originates from lattice sites with an electron-exciton double occupancy ($X_2$). Finally, we consider the case where the electronic doping is below the threshold required to reach a fermionic Mott insulator ($0 < \nu_e < 1$) but $I$ is strong enough to optically saturate the single-occupancy states. The extra excitons create a number of sites with electron-exciton or exciton–exciton double occupancies (panel f). In this case, the PL emission corresponds to mixed contributions from exciton–exciton and exciton–electron interaction ($U_{ex\text{-}ex}$ and $U_{ex\text{-}e}$); the individual peaks cannot be distinguished in a single spectrum due to the broadness of linewidths. Therefore, in this regime, the emitted light is only a combination of the $X_1$ and $X_2$ PL emission. This interplay between exciton and electron occupancy can lead to situations in which the moiré lattice is completely filled with a mixed population of fermions and bosons, forming a hybrid incompressible state. Specifically, in the limit of weak electronic tunneling, excitons can form a Mott insulating state, in the remainder of sites that are not filled by electronic doping. Note the line in Fig. 1c denoting panel f is an asymptote since optical pumping can not fully saturate an exciton line. At $\nu_e = 0$, this intensity is denoted as $I^*$ (see Supplementary Note 8 for details).

### System's properties for varying electronic (fermionic) occupation
To experimentally investigate these regimes, we perform PL measurements, with varying pump power and backgate voltage. A detailed description of the optical setup can be found in Supplementary Note 2.

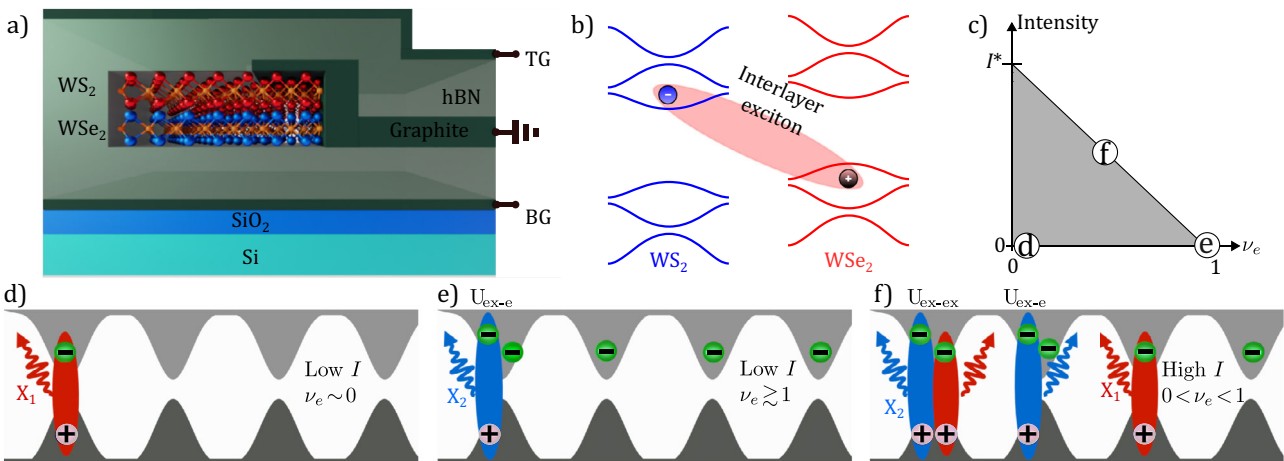

**Fig. 1 | $WS_2/WSe_2$ bilayer as a platform for correlated physics. a** Schematic of the $WS_2/WSe_2$ dual-gate device. The TMD heterobilayer is embedded between two symmetric gates: top gate (TG) and bottom gate (BG). **b** Depiction of the type-II band alignment of the bilayer. The blue and red curves denote bands from $WS_2$ and $WSe_2$, respectively. The shaded ellipse indicates the formation of interlayer excitons composed of an electron from the $WS_2$ conduction band and a hole from the $WSe_2$ valence band. **c** Phase diagram of the system. The population of the moiré lattice can be controlled via two independent parameters: the gate voltage changes the electronic filling factor ($\nu_e$), and the optical pump creates a population of excitons, proportional to the input intensity. In the gray area, the system behaves as a mixed gas of bosonic and fermionic particles. As one approaches the upper limit (black line), the system becomes incompressible due to the saturation of single-occupancy states. **d–f** Interlayer exciton formation under optical excitation for three different regimes governed by the pump intensity ($I$) and $\nu_e$: **c** low $I$ and $\nu_e$ ~ 0, **d** low $I$ and $\nu_e$ ~ 1, **e** high $I$ and $0 < \nu_e < 1$. $X_1$ ($X_2$) denotes PL emission from singly (doubly) occupied moiré lattice sites. $X_2$ can originate from either electron-exciton ($U_{ex\text{-}e}$) or exciton–exciton ($U_{ex\text{-}ex}$) double occupancies.

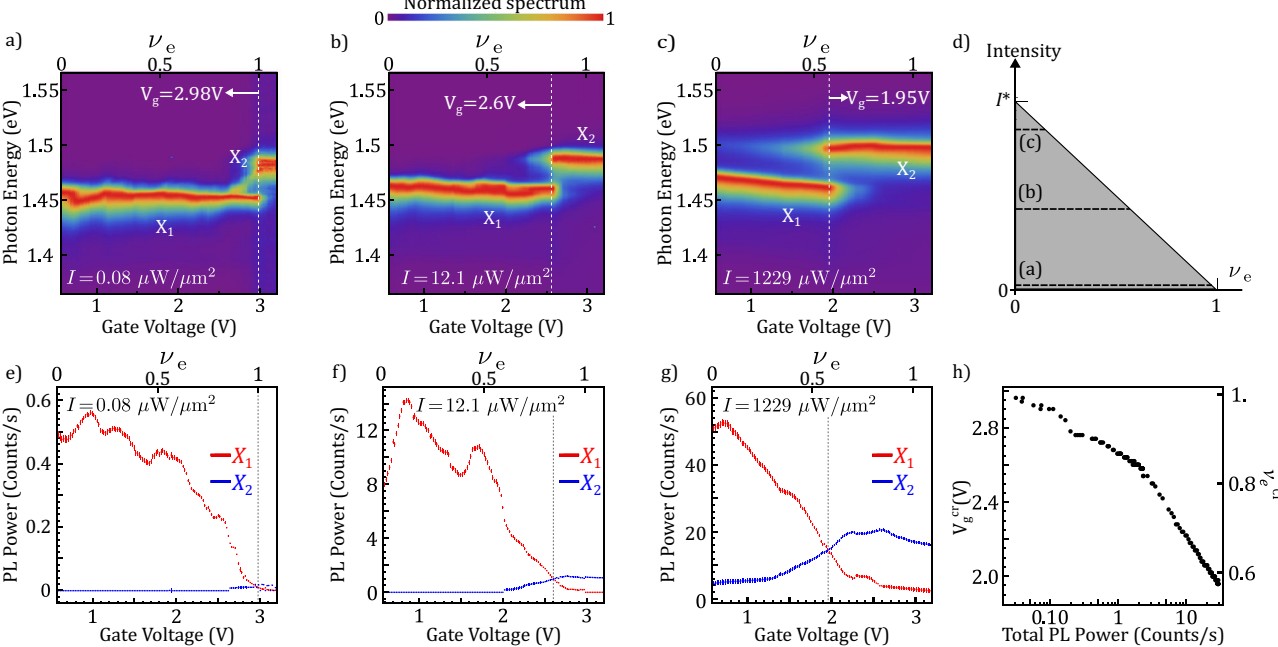

**Fig. 2 | System's properties for increasing electronic (fermionic) occupation.**
**a–c** Normalized PL spectrum as a function of gate voltage ($\nu_e$) for three different
pump intensities: $I = 0.08\,\mu W \mu m^2$ (**a**), $I = 12.1\,\mu W/\mu m^2$ (**b**) and $I = 1229\,\mu W/\mu m^2$ (**c**).
The peaks associated with single ($X_1$) and double ($X_2$) occupancy are indicated on
each panel. The dashed lines indicate the gate voltages at which the PL intensity $X_2$
exceeds $X_1$. The dashed black lines of (**d**) indicate the measurement ranges of (**a–c**).

**e–g** Evolution of the PL intensity for $X_1$ (red) and $X_2$ (blue) as a function of gate
voltage for the same values of pump intensities displayed in (**a–c**). The electron
filling factor at which $X_2$ exceeds $X_1$ decreases as pump intensity increases. **h** shows
the gate voltage at which the intensity of $X_2$ exceeds that of $X_1$, as a function of the
total PL intensity. The error bars represent the standard errors for the parameter
estimates in the fitting routine.

We use pulsed excitation to achieve high exciton density while reducing thermal effects by keeping low average power. Experiments with CW excitation are consistent with the presented data, as shown in Supplementary Note 6. Figure 2a–c shows the PL dependence at three different intensities as schematically shown in panel d. Figure 2a shows the normalized doping-dependent PL spectrum for low/(0.08 μW/μm²), which corresponds to low bosonic occupation. The fermionic occupation $\nu_e$ is varied between 0 and 1.1. For low $\nu_e$, PL emission is detected only from $X_1$. However, at $V_g \approx 2.98$ V, we detect a transition in the PL emission to $X_2$. This transition corresponds to the formation of X's in the presence of an incompressible fermionic Mott insulator[24,25]. From the reflectivity measurement and calculations from a capacitor model, we attribute $V_g = 2.98$ V to $\nu_e = 1$ (see Supplementary Note 3). The energy gap between $X_1$ and $X_2$ is $\Delta E \approx 29$ meV, which corresponds to the on-site Coulomb repulsion energy between an electron and an exciton ($U_{ex-e}$). We elaborate on this energy gap later in the sub-section "Energy map along the phase space". The dim mid-gap features between $X_1$ and $X_2$ at $\nu_e$ -1 are strongly position-dependent and disappear at higher power. This indicates that such emission is from localized excitons. Figure 2b shows the PL spectrum under pump intensity equal to 12.1 μW/μm². It is worth noticing that the $V_g$ at which the PL signal from $X_2$ is detected is lower than in panel a. The system is therefore in the regime depicted in Fig. 1f. Upon further increasing the pump intensity $X_2$ can be detected even at $\nu_e = 0$, as observed in Fig. 2c. In this case, the PL emission originates solely from double occupancy of excitons in a moiré lattice site, suggesting that, for high $I$, purely bosonic states of strongly interacting excitons are created. Comparing Fig. 2a, c, one can observe that in the former case, the emergence of the $X_2$ peak corresponds to a sharp suppression of $X_1$, while in the latter case, both peaks coexist. This indicates the nature of the double occupancy: in the first scenario, the exciton is forming in the presence of an electron, and after its recombination, there are no other optical excitations in the system. In contrast, the coexistence of both peaks in

panel c shows that upon double exciton occupancy, the recombination of $X_2$ precedes the recombination of $X_1$.

From the observation described in the previous paragraph, we conclude that the detection of PL emission with $X_1$ and $X_2$ energies benchmarks the formation of exciton states in singly and doubly occupied lattice sites, respectively. At $\nu_e = 0$, the $X_1$ peak in Fig. 2c is blueshifted with respect to Fig. 2a. We associate this feature with a mean-field effect due to exciton–exciton interaction. As we increase the electronic doping, fewer sites are available to create $X_1$ excitons and on those occupied sites, only $X_2$ is created. Consequently, the effective population of $X_1$ excitons is decreased. Therefore the mean-field shift is suppressed to the point that at high filling ($\nu_e$ -1) the $X_1$ energy is the same as in the case of low pump intensity. Next, in order to understand the interplay between fermionic and bosonic lattice occupancies in each regime, we perform a quantitative analysis of their respective integrated intensity. We extract these values from the collected PL spectra using a computational fitting method (see Supplementary Note 7 for further details). Figure 2e–g displays this intensity dependence on $\nu_e$ for the same $I$ range of panels a–c. We notice that as electrons fill the system's phase space (upon increasing $V_g$), the number of accessible single-occupancy states decreases. As a consequence, the integrated intensity of $X_1$ reduces with increasing $\nu_e$. Remarkably, for each intensity, there is a critical $\nu_e$ after which the PL emission of $X_2$ exceeds that of $X_1$. The gate voltage at which the crossing takes place ($V_g^{cr}$) is highlighted on each panel by a vertical dashed line. This line indicates a constant ratio between the $X_1$ and $X_2$ populations. The crossing takes place at lower $\nu_e$ upon increasing $I$, as expected. In Fig. 2h, we track $V_g^{cr}$ as a function of the total collected PL emission, which gives an indication of the total number of excitons in both $X_1$ and $X_2$ branches. We observe a clear trend: a higher total population of excitons results in a faster saturation of the single-occupancy states and hence an increasing number of double occupancy states.

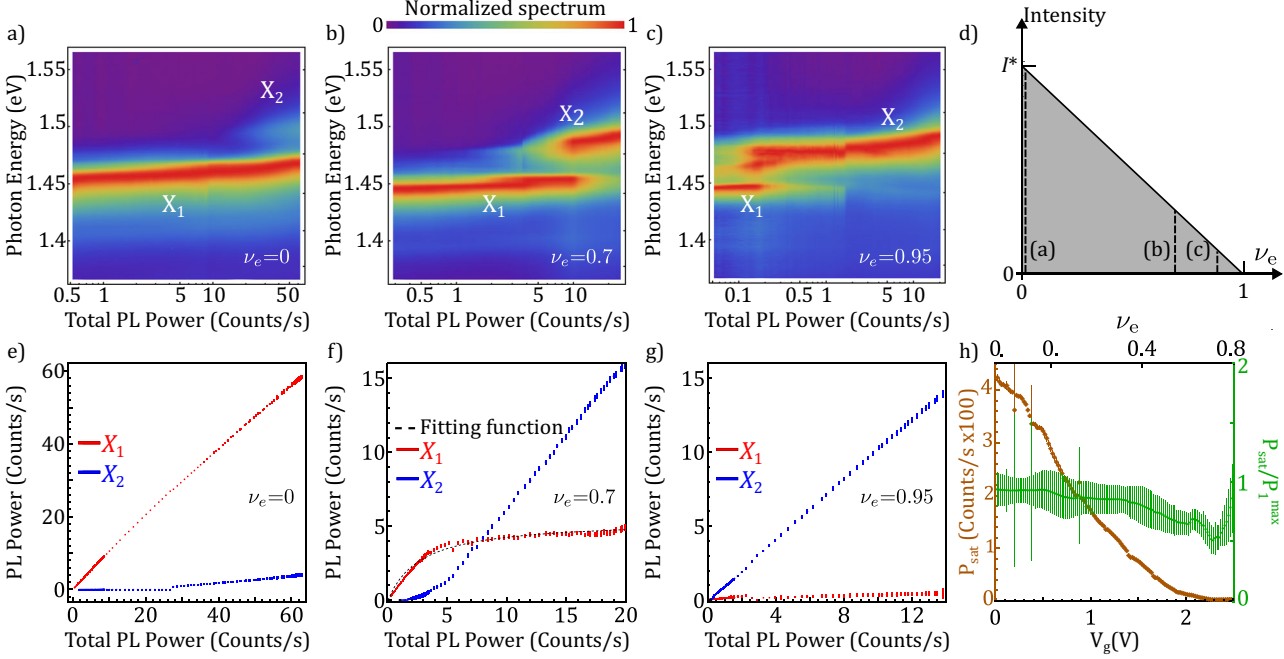

**Fig. 3 | System's properties for increasing excitonic (bosonic) occupation.**
**a–c** Normalized PL spectrum as a function of the total collected PL power for three different electronic filling factors. The peaks associated with single ($X_1$) and double ($X_2$) occupancy are indicated on each panel. **d** indicates the ranges of $I$ and $\nu_e$ for the measurements shown in (**a–c**). **e–g** evolution of the PL power for $X_1$ (red) and $X_2$ (blue) as a function of the total collected PL power for the same values of $\nu_e$ displayed in (**a–c**). **f** displays the fitting function (dashed black line) employed to extract $P_{sat}$ and $P_1^{max}$ (described in the text). **h** Evolution of $P_{sat}$ (brown) as a function of the gate voltage ($\nu_e$). As expected from our phase-space filling model, its value reduces with increasing filling factor. The quantity $P_{sat}/P_1^{max}$ (green) shows good agreement with the theoretical model. The error bars represent the standard errors for the parameter estimates in the fitting routine.

## System's properties for varying excitonic (bosonic) occupation

Next, in order to trace the role of the optical pump and the optical saturation that leads to the formation of incompressible bosonic states, we investigate the PL for varying $I$ for different $\nu_e$. In Fig. 3a–c, we focus on three different values of $\nu_e$, as indicated in panel d, and study the PL spectrum for increasing emitted PL power. For zero fermionic occupancy (panel a), $X_2$ contributes to the emission only at very high total PL emission intensity. In panels b and c, we increase the electronic doping to $\nu_e = 0.7$ and $\nu_e = 0.95$, respectively, and as expected, the total PL at which we detect $X_2$ decreases. In the low-power region, panel c shows the PL emission from mid-gap states also observed in Fig. 2a. Apart from the energy gap in the emission, we observe a blueshift of the $X_1$ line with increasing PL power. Assuming the weak tunneling regime, this shift should be equal to $U_{ex-ex}\langle \hat{x}^\dagger \hat{x}\rangle$, where $\hat{x}^\dagger$ is the creation operator of an exciton. For example, in Fig. 3b for total PL power at 2 counts/s, the bosonic occupation is $\langle \hat{x}^\dagger \hat{x}\rangle \simeq 0.2$. This corroborates with the energy gap that occurs at 10 counts/s for an estimated unity filling ($\langle \hat{x}^\dagger \hat{x}\rangle \simeq 1$). We present a fully quantum theoretical analysis of this observation in Supplementary Note 10. Panels e–g show the intensities of $X_1$ and $X_2$ for the values of $\nu_e$ in panels a–c. As expected, in panel e, one can observe that the intensity of the $X_1$ PL emission increases monotonically, and it starts to saturate only at very high total PL emission regimes. Upon filling the moiré lattice with one exciton or one electron per site, the $X_1$ PL intensity saturates. With higher $\nu_e$, the saturation occurs at lower $I$, as shown in panels f and g. Since this saturation corresponds to filling the single-occupancy states, we associate it with the establishment of an incompressible bosonic Mott insulator. Note that this bosonic Mott insulator is in a drive-dissipative regime, similar to the demonstration in superconducting qubit systems[26].

To quantitatively analyze this saturation effect, we fit the $X_1$ PL power ($P_1$) to the function $P_1 = P_1^{max} \frac{P}{P + P_{sat}}$, where $P$ is the total PL

power. From the fitting, we extract $P_1^{max}$ which is the asymptotic value of the $X_1$ emitted PL power, and $P_{sat}$ which determines the total PL of saturation. This functional form corresponds to the expected system behavior when the charge gap U is sufficiently large to permit the utilization of a phase-space filling model to treat both single and double occupancy states (details in Supplementary Note 8). Figure 3f includes an example of the fitting function (dashed black line). According to our model, the value of $P_{sat}$ should decrease with increasing $\nu_e$ because a lower excitonic population is required to achieve the incompressible states. The compiled data for the full range of $\nu_e$, shown in panel h with brown marks, is in good agreement with the expected trend. From the model, we can also infer that the quantity $P_{sat}/P_1^{max}$ should be independent of the electronic doping level because both quantities depend linearly on $1 - \nu_e$; higher electronic occupancy implies less single-occupancy states available to host an exciton. The green marks in Fig. 3h represent this behavior, which is in good agreement with the model. We conclude that the saturation of single-occupancy states is directly reflected in the intensity of $X_1$, enabling the extraction of the conditions under which the incompressible states occur. Importantly, this enables a direct calibration of the bosonic and fermionic fractions in the system.

## Exciton diffusion measurements

In order to further validate the incompressible nature of excitonic states, we perform diffusion measurements of the interlayer excitons[27]. For a steady population of excitons created by a continuous-wave laser pump, the diffusion length carries information about the nature of the state: an incompressible bosonic state is expected to have a lower diffusion length than a weakly interacting one. We spatially image the diffusion pattern with spectral resolution and extract the diffusion

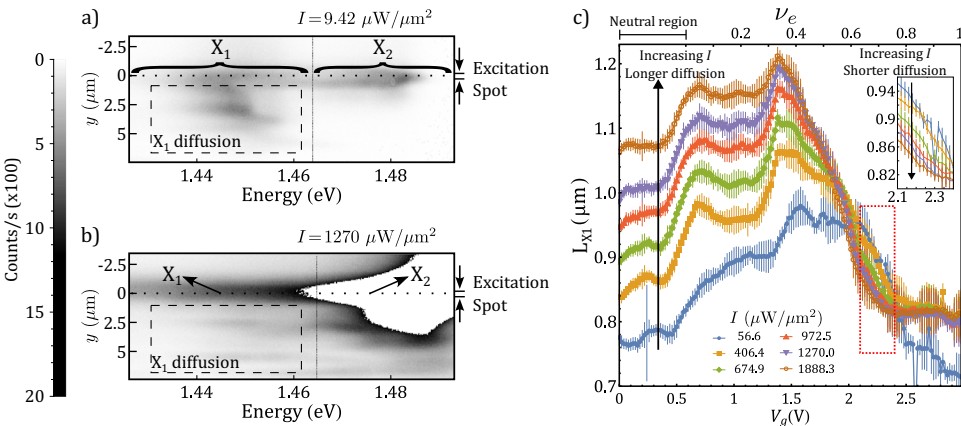

**Fig. 4 | Exciton diffusion and incompressibility.** Spectrally and spatially resolved diffusion pattern at $\nu_e = 0.73(V_g = 2.34V)$ for low (**a**) and high (**b**) $I$. The dashed rectangle highlights the region where the suppression of diffusion can be observed. **c** Exciton diffusion length as a function of the gate voltage for a range of $\nu_e$ and for different input intensities. For low $\nu_e$, the diffusion length increases with $I$ due to

exciton repulsive interaction. Upon further filling the moiré lattice, the trend inverts, indicating the optical realization of incompressible states. The inset is a zoom-in of the red dotted rectangle to highlight the reduction of $L_{X1}$ with increasing $I$. The error bars represent the standard errors for the diffusion length estimated from the exponential fitting.

length ($L_{X1}$) of the single-occupancy excitons. The choice of $L_{X1}$ as an appropriate quantity to benchmark the incompressibility of bosonic Mott insulating states, assumes a constant exciton lifetime with varying population. This is supported by previous reports in the literature that show the independence of this quantity over three orders of magnitude of pumping power[28]. The downward diffusion image has patterns that originate in the inhomogeneous surface of the bilayer. Although the inhomogeneities on that side hinder the extraction of $L_{X1}$, the optically induced suppression of the diffusion length for constant $\nu_e$ can be clearly observed in this region (Fig. 4a, b). The population injected at $y = 0$ (dotted line) propagates, and its emission pattern is monitored along a range of 5 μm (dashed rectangle). The color scale is the same for both panels. Panel b shows a reduction of the diffused $X_1$ population in comparison to panel a. For the quantitative analysis of this observation, it is necessary to use a fitting routine, for which the smooth pattern on top of the injection point ($y < 0$) is more reliable. Figure 4c shows the extracted $L_{X1}$ as a function of $V_g$ for different pump intensities from the exponentially decaying spatial diffusion pattern in this region. We provide more details about the analysis of the diffusion data in Supplementary Note 9. For low electronic density, the exciton diffusion length increases as the power is augmented. This trend, highlighted by the upward arrow, is in agreement with the expected behavior for weakly interacting bosons[28,29]. Remarkably, as the electronic filling factor increases, the trend completely inverts (inset). This is a direct signature of the bosonic Mott insulator formation since the bulk is incompressible and the melting only occurs at the edge.

### Energy map along the phase space

The implemented fitting algorithm allows us to track the changes in the energy of both species of excitons and the energy gap between them. These results are presented in Fig. 5. Panels a and b show the central energies of the peaks $X_1$ and $X_2$ in the space of parameters for which each peak is detectable. In the range where both of them can be detected, their energy difference $\Delta E$ (panel c) provides important information about the nature of the interactions taking place in the system. In the case of low electronic occupancy and high exciton density (top left corner of the panel), $\Delta E$ corresponds to the exciton–exciton interaction gap (Uex-ex ~32 meV). Conversely, at high $\nu_e$ and low exciton density (bottom right corner), this gap depends on the exciton–electron interaction (Uex-e ~27 meV). The gradual change in the nature of the interactions taking place in the system along the parameters space is reflected in the change of $\Delta E$.

Interestingly, the largest energy gap takes place for states with high occupation of bosons and fermions (top right corner), which is consistent with a blueshift of the $X_2$ PL peak due to the high population of excitons with large Bohr radius repelling through dipolar interaction.

## Discussion

In summary, we demonstrated a Mott insulating state of excitons in a hybrid Bose–Fermi Hubbard system formed in a TMD heterobilayer. While our incompressibility observation was based on spatially resolved diffusion in the steady-state limit, one can explore interesting non-equilibrium physics due to the relatively long lifetime of interlayer excitons. More generally, spatiotemporally resolved measurements, combined with independent tunability of fermionic and bosonic populations, make it possible to investigate the equilibrium and non-equilibrium physics of Bose–Fermi mixtures. Moreover, a quantum microscopic model capable of fully describing such a driven-dissipative Bose–Fermi mixture remains an open area of research. The novel experimental diffusion method used to benchmark the excitonic incompressibility opens exciting perspectives for the simulation of complex dynamics in many-body quantum systems that range from a single bosonic particle in a Fermi sea to a strongly interacting gas of bosons. Particularly intriguing examples are the optical investigation of charge and spin physics in integer and fractional fillings, e.g., Mott excitons[30,31] or spin liquids[32–35], and fractional Chern insulators[36,37].

## Methods

### Device fabrication

The WSe$_2$/WS$_2$ heterostructure was fabricated using a dry-transfer method with a stamp made of a poly(bisphenol A carbonate) (PC) layer on polydimethylsiloxane (PDMS). All flakes were exfoliated from bulk crystals onto Si/SiO$_2$ (285 nm) and identified by their optical contrast. The top/bottom gates and TMD contact are made of few-layer graphene. The PC stamp and samples were heated to 60 °C during the pick-up steps and released from the stamp to the substrate at 180 °C. The PC residue on the device was removed in chloroform, followed by a rinse in isopropyl alcohol and ozone clean. Sample transfer was performed in an argon-filled glovebox for improved interface quality. The electrodes consist of 3.5 nm of chromium and 70 nm of gold. They were fabricated using standard electron-beam lithography techniques and thermal evaporation.

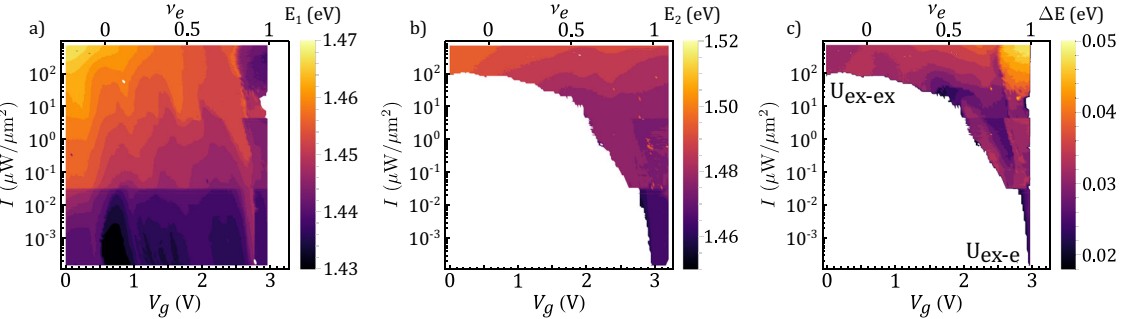

**Fig. 5 | X$_1$ and X$_2$ exciton energies along the phase diagram.** Energy of the X$_1$ (**a**) and X$_2$ (**b**) PL emission as a function of gate voltage and pump intensities. The white areas correspond to the range of parameters where the corresponding peak completely vanishes. When X$_1$ and X$_2$ coexist, we extract the energy difference, as shown in (**c**).

## Optical measurements

The sample is kept in a dilution refrigerator at a temperature of 3.5 K. For PL measurements, we use a confocal microscopy setup. Our pumping source is a pulsed Ti:Sapphire laser tuned at 720 nm (1.722 eV), with a pulse duration of 100 fs and a repetition rate of ~80 MHz. In addition, an optical chopper system at 800 Hz is used to prevent sample heating while having a high pump intensity. The residual pump is removed with a spectral filter before collecting the PL emission in a spectrometer-CCD camera device. A complete description of the setup is presented in the Supplementary Note 2.

For the diffusion measurements, we used a continuous-wave (CW) laser. The rest of the optical measurement setup was similar.

## Data availability

The PL and diffusion data generated in this study have been deposited in the Figshare database under accession links: https://doi.org/10.6084/m9.figshare.25246012.v1; https://doi.org/10.6084/m9.figshare.25246006.v1; https://doi.org/10.6084/m9.figshare.25246009.v1; https://doi.org/10.6084/m9.figshare.25246015.v1 .

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

## Acknowledgements

The authors acknowledge fruitful discussions with N. Schine and A. Kollar. This work was supported by AFOSR FA95502010223, MURI FA9550-19-1-0399, FA9550-22-1-0339, NSF IMOD DMR-2019444, ARL W911NF1920181, and Simons and Minta Martin foundations. Ming Xie is supported by the Laboratory for Physical Sciences. R. Ni and Y. Zhou are supported by the U.S. Department of Energy, Office of Science, Office of Basic Energy Sciences Early Career Research Program under Award No. DE-SC-0022885.

## Author contributions

B.G., D.G.S.F., S.S. and M.H. conceived and designed the experiments. K.W. and T.T. supplied the necessary material for the fabrication of the sample. B.G., D.S. and R.N. designed and fabricated the sample. J.V. and S.M. collaborated with the preparation of the setup at its initial stage. B.G., D.G.S.F. and S.S. performed the experiments. B.G., D.G.S.F., S.S., T.S.H., M.J.M., M.X., A.I., Y.Z. and M.H. analyzed the data and interpreted the results. T.S.H. and M.H. elaborated on the theoretical models presented in the manuscript. B.G., D.G.S.F., S.S., M.J.M. and M.H. wrote the manuscript, with input from all authors.

## Competing interests

The authors declare no competing interests.
