## [Peer Review File · Nature Communications]

REVIEWER COMMENTS

Reviewer #1 (Remarks to the Author):

Gao et al. applied optical spectroscopy to study the WS₂/WSe₂ heterobilayer. The author first used a pulsed laser (pulse duration 100 fs) as excitation to study the PL spectra as a function of electron doping. It is found that a new PL peak would show up at increased excitation power at zero doping, while the onset excitation power is decreased for finite doping. The authors believe that the optical excitation power control the exciton density. The authors proposed that when the exciton density plus electron density is equal to the moire superlattice density, the mixture of excitons and electrons forms a new Mott insulator state. The authors then used a CW laser to study the diffusion of the two types of excitons through PL measurements. It is found that the diffusion length decreases after certain electron doping, which the authors attributed to the signature of an incompressible state.

The UCSB group has studied the 60-degree twisted WS₂/WSe₂ heterobilayer using a differential PL technique that can probe the incompressibility. They found both the exciton Mott and electron-exciton hybrid Mott state, with the $U_{e-ex} \sim 35$ meV and $U_{e-e} \sim 15$ meV. The UCSB has also combined PL and absorption spectra to generate the phase diagram for the hybrid Mott state and exciton Mott state. The work from the UCSB group has been posted on arxiv for about a year and has recently come out in May [SCIENCE 380, 860-864 (2023)].

For this paper, I am concerned with the experimental design and also the majority of the data analysis. The TMD heterobilayer is a rich system for correlated physics. So careful design and analysis are needed to extract useful information. In the context of this work, the interlayer excitons have a finite lifetime and are not as static as the electrons. In the pulse excitation experiment, the extremely short pulse will generate a large population of excitons in a short time, and the system likely starts from a highly nonlinear and nonequilibrium state. On the other side, the CW excitation will constantly generate an exciton density linearly at small excitation power. These two completely different excitations might probe different physics of the same system. The authors need to justify their experimental design. I also found many analyses are based on bold assumptions that are likely unfounded. I would not recommend publishing this work in Nature Communications.

I also include my specific comments here.

1) Why didn't the authors perform the first half of the experiments using the CW excitation, which seems to be more suitable for this study? Could the observations in Fig. 2 be a result of photodoping of the pulsed laser? The authors could not resolve the difference between U_{e-ex} and U_{e-e} , while the

difference can be as large as 20 meV in the UCSB report. The authors observed a shift of 29 meV, close to the Ue-ex reported in the UCSB report (32-35 meV). Could the authors be seeing the Ue-ex? That would be consistent with the photodoping picture.

2) The authors should show the results from the hole doping. The holes should also form a hybrid Mott state with excitons.

3) For Fig. 2a, even if the excitons form a hybrid Mott state with electrons, and the next exciton has to occupy higher energy and emit light there (X2), the exciton on the vacant moire superlattice is not going to live forever and will recombine and give out photon at the energy of X1. Why would X1 PL stop? Also, what is the additional feature near $\nu_e=1$? There is an additional PL before $\nu_e=1$.

4) The PL analysis is troublesome. The authors claim "assuming that the radiative decay rate of both lines is similar." This is a bold claim. What evidence do the authors have to support that? Did the authors measure the lifetime? The higher energy PL often has a shorter lifetime and probably a lower quantum yield.

5) The authors also claim, "We notice that as electrons gradually fill the system (upon increasing V_g), the number of accessible single-occupancy states decreases. As a consequence, the integrated intensity of X1 reduces with increasing ν_e ." The authors seem to picture the added electron occupying each site sequentially. If so, this picture is wrong. The electron is, in general, delocalized before the phase transition happens. The decrease in PL can be simply attributed to the increased nonradiative channels to the free electrons.

6) There are two peaks in X2 in Fig. 3c that the authors did not discuss.

7) The diffusion pattern in Fig. 4a is weird, with PL intensity fluctuating across y . Is that spatial inhomogeneity? How could diffusion length be extracted with such a pattern?

8) The authors claim that "an incompressible bosonic state is expected to have a lower diffusion length than a weakly interacting one." Shouldn't the diffusion constant be more relevant to probe the compressibility?

9) The authors claim diffusion is suppressed at higher excitation power in Fig. 4b. But this suppression could be just due to the shortened lifetime at high excitation power. That is not necessarily a sign of incompressibility. Have the authors measured the lifetime for different excitation power?

10) I cannot tell what the trend at $\nu=0.6$ the author talked about is. Could the author elaborate on that?

11) An important question about diffusion fitting: How do the authors know the linear rate equation is suitable across more than two orders of magnitude increase of excitation power? Have the authors measured the excitation power dependence of the PL? Is that a linear function?

12) There is no discussion of the entire Fig. 5 in the main text.

13) What is the twist angle between WSe₂ and WS₂? Did the author reproduce their results in other samples?

Reviewer #2 (Remarks to the Author):

The manuscript report experimental measurements of the WS₂/WSe₂ heterobilayer with changing the electron density and exciton density (through the intensity of optical pump). The major discovery is a 'Mott insulating' state when the total density of electron and exciton is 1.

The measurement of the exciton density seems to be indirect, because there is no clear quantitative relationship between the optical power and exciton density. Also the major evidence of the "Mott gap" is through the change of the energy in PL spectrum, without direct transport or charge compressibility measurement. But the interpretation looks promising to me, and I think the paper is suitable for natural communication.

I have a few comments/suggestions which I hope the authors could consider:

(1) The definition of the filling ν_e is not very clear. I need to guess that it means the electron density in the conduction band, but from WS₂ or WSe₂?

(2) I'm not sure the word 'bosonic Mott insulator' is a good terminology. In my understanding, the phase refers to an "insulator" with the total number of electrons and exciton is 1 per moire site. So it is apparently not purely 'bosonic'.

Reviewer #3 (Remarks to the Author):

“Excitonic Mott insulator in a Bose-Fermi-Hubbard system of moire WS₂/WSe₂ heterobilayer” authored by Gao, etc. studies the correlation between excitonic states and free carriers in TMDC heterostructure, demonstrated exciton Mott insulator states, which is a hot topic. The work agrees with previously reported results and provides new insight through the spatially resolved measurements. I would support publishing the manuscript with some edits. Also, there are some questions the authors may try to address in the revision.

1. Fig 4 shows the spatially resolved measurement. In fig 4a and 4b, the spatial distribution seems to exhibit some features (horizontal stripes). Are those features real or they are artifacts from the measurements? If it is real, could the authors comment on the feature? Also, as the excitons diffuse away from the pumping region, the density will decrease dramatically and may have different diffusion constants. I am curious whether it will impact the analysis of the diffusion length. Also, as the exciton density decreases, shall we expect a transition from the X₂ exciton to the X₁ as moving away from the pumping region?

2. In fig 4c, the variation of the X₁ diffusion length seems to have a similar pattern. After the electron doping reaches ~ 0.1 filling, the diffusion lengths decrease and then hit a plateau. When electron filling reaches ~ 0.3 it starts to increase again. Is there any explanation for this phenomenon? Could it be possible to have some intermedia state at the plateau region, for example, correlated electron states freeze the exciton? Also, the diffusion lengths vs doping curves are very similar in the low doping region except for the overall increase of diffusion length with the increasing exciton injection. The overall increase of diffusion length seems to suggest ex-ex interaction is strong enough to change the diffusion length. Meanwhile, the doping-induced X₁ to X₂ transition seems ex-e interaction

significantly changes the available site for single occupied excitons. One may expect the transition features of diffusion length will shift to lower electron doping levels at higher pumping densities. The author may add the X1 to X2 transition voltages for the different pumping power in figure 4c to help readers to interpret the transitions.

3. In Fig 2 as the doping increases the X1 excitons exhibit an energy redshift. Such phenomena are more obvious in the high pumping power region. Is there an explanation for the shift? Is the shift from electron-exciton interaction or the interaction of the exciton dipole with the applied gate voltage?

4. It is inevitable to compare the result of the manuscript to ref 36, which studies similar physics in the same material system. It seems that the pumping density (exciton injection rate) is quite different between the manuscript and ref. 36 (CW laser pump). Could the authors calibrate exciton density to compare the results? Also, did the author lock the CCD to the pumping pulse? If not, will the dynamics of the excitons impact the result? For example, after the pump pulse, as the excitons recombine and diffuse away from the pumping region, the overall exciton density will decrease. If start within the X2 region, would it lead to a transition from X2 to X1? If the CCD keeps integrating PL signal, and such transition exists, would it be responsible for the long tails of X1 and X2 (for example fig 2c, fig. 3b,c)?

5. In ref 36, they divide the phase diagram into 3 regions and separated the exciton-exciton interaction from the exciton-electron interaction. The authors claimed, "within our experimental resolution, these latter two situations are not spectrally resolvable." However, fig.5c indicates that the ex-ex and ex-e states can be separated by comparing X1 X2 energy as well as the filling numbers of exciton and electron. Could the authors clarify their argument? The definition of ex-ex interaction may need to be clarified as well. When authors discuss the blueshift of X1 exciton energy in pumping power dependence measurement (fig3), U_{ex-ex} is quoted. I would assume this is the interaction between excitons at different sites. However, in fig 5c, the U_{ex-ex} seems to be ex-ex interaction of exciton from the same site. Also, there is no discussion about fig. 5. in the manuscript.

6. Since it seems to be possible to separate the ex-ex, ex-e states. Could authors comment on the X2 diffusion data in SI and whether any feature could be correlated to the two different states?

In conclusion, the manuscript presents high-quality experimental data. The correlation between excitonic states and free carriers in the 2D TMDC heterostructure is of broad interest and is under heavy investigation. The main conclusion agrees with recent studies on the same topic. More discussion about the novel spatially resolved measurement could further improve the manuscript.

Response letter to reviewers

Reviewer #1 (Remarks to the Author):

Gao et al. applied optical spectroscopy to study the WS₂/WSe₂ heterobilayer. The author first used a pulsed laser (pulse duration 100 fs) as excitation to study the PL spectra as a function of electron doping. It is found that a new PL peak would show up at increased excitation power at zero doping, while the onset excitation power is decreased for finite doping. The authors believe that the optical excitation power control the exciton density. The authors proposed that when the exciton density plus electron density is equal to the moire superlattice density, the mixture of excitons and electrons forms a new Mott insulator state. The authors then used a CW laser to study the diffusion of the two types of excitons through PL measurements. It is found that the diffusion length decreases after certain electron doping, which the authors attributed to the signature of an incompressible state.

The UCSB group has studied the 60-degree twisted WS₂/WSe₂ heterobilayer using a differential PL technique that can probe the incompressibility. They found both the exciton Mott and electron-exciton hybrid Mott state, with the $U_{\text{ex-e}} \sim 35$ meV and $U_{\text{e-e}} \sim 15$ meV. The UCSB has also combined PL and absorption spectra to generate the phase diagram for the hybrid Mott state and exciton Mott state. The work from the UCSB group has been posted on arxiv for about a year and has recently come out in May [SCIENCE 380, 860-864 (2023)].

For this paper, I am concerned with the experimental design and also the majority of the data analysis. The TMD heterobilayer is a rich system for correlated physics. So careful design and analysis are needed to extract useful information. In the context of this work, the interlayer excitons have a finite lifetime and are not as static as the electrons. In the pulse excitation experiment, the extremely short pulse will generate a large population of excitons in a short time, and the system likely starts from a highly nonlinear and nonequilibrium state. On the other side, the CW excitation will constantly generate an exciton density linearly at small excitation power. These two completely different excitations might probe different physics of the same system. The authors need to justify their experimental design. I also found many analyses are based on bold assumptions that are likely unfounded. I would not recommend publishing this work in Nature Communications.

We appreciate the reviewer's thorough assessment of our work and we took very seriously their concerns regarding the experimental design and data analysis. As mentioned by the reviewer, the TMD heterobilayer system is acquiring increasing significance as a platform to study matter correlated states. We also agree with the fundamental importance of accurate experimental designs and analysis to probe these physics. In response to the specific concerns raised, we would like to provide the following clarifications and justifications:

Choice of Excitation Methods: The use of both pulsed and CW excitation methods was deliberate and served specific purposes in our study. The pulsed excitation allows us to reach high excitonic densities with low average power, something critical to avoid slow thermal effects in the system. The use of CW excitation, however, is mandatory to study the steady state of the system, which was necessary for the diffusion measurement. This is further discussed in the next point.

We are aware of the works from UCSB (Xiong et. al., Science 380,860-864 (2023)) and UW (Park et.al., Nature Physics (2023)) in similar systems, as the disclaimer in our manuscript indicates. The paper from UCSB was published on Arxiv in July 2022, when our work was at an already advanced stage, and we initially submitted our manuscript in March 2023, when this work had not been published in any peer review journal. Several weeks after our submission, the two mentioned papers were published. We would like to emphasize the relevance of these works as well as highlight that our contribution includes: original experimental methods, e.g. diffusion experiments to study incompressible states in the system, and an original theoretical driven-dissipative model that provides deep understanding of the observations beyond the descriptions in the published manuscripts. The contributions from UCSB and UW will be properly included and acknowledged in an eventual published version of our manuscript, for which we will refer to the editor to do it in the most appropriate way. Additionally, we are aware of a recently published pre-print that also studies the physics of excitonic Mott insulating states in a similar platform (Lian et. al., arXiv:2308.10799v2), something that we also interpret as a manifestation of the high relevance and timeliness of this research topic.

We understand the referee's concern about the assumptions behind our conclusions, which we will address in detail in the following points.

I also include my specific comments here.

1) Why didn't the authors perform the first half of the experiments using the CW excitation, which seems to be more suitable for this study? Could the observations in Fig. 2 be a result of photodoping of the pulsed laser? The authors could not resolve the difference between $U_{\text{ex-e}}$ and $U_{\text{e-e}}$, while the difference can be as large as 20 meV in the UCSB report. The authors observed a shift of 29 meV, close to the $U_{\text{ex-e}}$ reported in the UCSB report (32-35 meV). Could the authors be seeing the $U_{\text{ex-e}}$? That would be consistent with the photodoping picture.

Although we performed experiments on both CW and pulsed regimes, the pulsed excitation has the important capability of reaching densely populated states while avoiding thermal effects. This is possible because the excitonic population depends on the peak power (the system has nanosecond dynamics) but the thermal effects are due to absorption of light from the substrate and depend on the average power. Motivated by the point raised by the reviewer, we collected additional data in the CW excitation regime to verify its consistency with the data presented in the paper. We used a Ti:Sapphire CW laser and performed the experiment in the same conditions. As shown in Fig. R1, the qualitative behavior of the system is the same as in Fig. 2

of the main text: at low gate voltage only the single occupancy states are detected, and for increasing exciton density a secondary peak indicates the creation of double occupancies in the moiré lattice. The gate voltage dependence is also consistent: as one populates the lattice with electrons (by increasing the filling factor in panels e and f), the secondary peak becomes visible at lower gate voltages. To clarify this point for the reader, we are adding this figure and its discussion to the Supplementary Material. In the main text, we also include the following sentence to clarify this point:

“We use pulsed excitation to achieve high exciton density while reducing thermal effects by keeping low average power. Experiments with CW excitation are consistent with the presented data, as shown in the Supplementary Material (section VI).”

Figure R1. Evolution of the normalized PL spectrum as a function of the total collected PL power for three different electronic filling factors under pulsed (a-c) and CW (d-f) excitations. The peaks associated with single (X1) and double (X2) occupancies are indicated on each panel. The qualitative behavior remains unaltered under both excitation regimes.

Regarding the difference between $U_{\text{ex-e}}$ and $U_{\text{ex-ex}}$, we realize that the sentence “...within our experimental resolution, these latter two situations (ex-ex or e-ex) are not spectrally resolvable.” (page 3) might be misleading. That statement is intended to remark on the fact that we never detect two resolvable peaks for X2 in a single spectrum. This is due to the broad linewidth of this emission with respect to the energy difference between $U_{\text{ex-e}}$ and $U_{\text{ex-ex}}$. The PL is composed of two peaks at different energies and with variable intensities. Effectively, this leads to a broad PL emission whose central energy is determined by the relative intensity of the two component peaks. However, in the two limiting cases of purely fermionic and bosonic fillings, the gap energy has a single origin: $U_{\text{ex-e}}$ and $U_{\text{ex-ex}}$, respectively. As shown in Fig. 5 (which we will further comment in point 12), the quantitative analysis of the PL data allows us to present these two

experimental values and all the intermediate cases. To clarify this in the manuscript, we are changing the referred statement as follows:

“In this case, the PL emission corresponds to mixed contributions from exciton-exciton and exciton-electron interaction ($U_{\text{ex-ex}}$ and $U_{\text{ex-e}}$); the individual peaks cannot be distinguished in a single spectrum due to the broadness of linewidths.”

The values of $U_{\text{ex-e}}$ and $U_{\text{ex-ex}}$ are determined to be 29 meV and 32 meV, respectively. This can be observed in Fig. 5.

In conclusion, the photo-doping picture mentioned by the reviewer is indeed the correct picture: increasing the pumping power increases the excitonic population in the moiré structure and changes the interactions taking place in the system. The change in the value of U that we show in Fig. 5, provides information about their nature, and it allows us to resolve $U_{\text{ex-ex}}$ and $U_{\text{ex-e}}$. It is worth noting that since the double electron occupancy does not lead to any optical signal, a measurement of $U_{\text{e-e}}$ would involve transport experiments beyond the scope of this work.

2) The authors should show the results from the hole doping. The holes should also form a hybrid Mott state with excitons.

Different works have concluded that at the hole-doped side, holes and excitons are localized in different high symmetry points of the moiré lattice, which hinders the possibility of achieving a hybrid exciton-hole Mott insulating state (Naik et. al., Nature, 609, 52–57 (2022), Lian et. al. arXiv:2308.10799v2). This statement is in agreement with the results of the UCSB paper; in the Supplementary Material, they claim: “We find that hole doping does not cooperate with excitons to form a mixed correlated insulator, and instead will destabilize the correlated insulator This can be potentially understood from the fact that the lowest energy conduction band orbital (CBM) and highest energy valence band orbital (VBM) are localized around different high symmetry points within a moiré unit cell, as recently reported by both theory and experiments.” The holes injected into the system do not cooperate with excitons in forming a lattice but rather act as an effective disorder that destabilizes it. In other words, the periodic potential observed by excitons is different from the one observed by holes. In Fig. S3 of the Supplementary Material, we show the PL response of the system upon a full sweep over the gate voltage. Notice how the gap on the hole-doping side is smaller than the one on the electron-doping side. This observation indicates that the exciton-hole interaction is weaker than the exciton-electron interaction, which justifies the choice of electronic doping for the study of hybrid Mott insulating states. This observation is not highlighted in the original version of the manuscript, for which we are adding the following paragraph to the new version of the Supplementary Material:

“Although this calibration holds for both the hole-doped and the electron-doped sides, Fig. S3b shows an asymmetric behavior of the exciton interaction with the sign of the doping: the energy gap at $\nu_e = 1$ on the electron doping side is larger than the one in the hole doping side. This is in agreement with theoretical predictions [Naik et. al., Nature, 609, 52–57 (2022)] and experimental observations [Xiong et. al., Science 380, 860-864 (2023), Lian et. al.

arXiv:2308.10799v2] that show a reduced value of the interaction strength due to the fact that holes and excitons reside in different high symmetry points of the moiré lattice. This observation indicates that electron doping is more suitable for the study of Fermi-Bose hybrid correlated states”

This modification will clarify for the reader the reason for choosing the electron doping side over the hole doping side for the study of hybrid correlated states.

3) For Fig. 2a, even if the excitons form a hybrid Mott state with electrons, and the next exciton has to occupy higher energy and emit light there (X2), the exciton on the vacant moire superlattice is not going to live forever and will recombine and give out photon at the energy of X1. Why would X1 PL stop? Also, what is the additional feature near $\nu_e=1$? There is an additional PL before $\nu_e=1$.

The PL associated with X2 can have two different physical origins: in the first case, it results from an exciton and an electron residing in the same moiré site. In this condition, only X2 can be detected through PL, because after the recombination, only the electron remains in the unit cell. In the second case, X2 arises from two excitons residing in the same moiré site, allowing both X1 and X2 to be detected through PL. These two scenarios are corresponding to the interaction energies U_{ex-e} and U_{ex-ex} , respectively. In the specific conditions depicted in Fig. 2a, the system operates in a very low-power regime, where the population of excitons is so low, that double exciton occupancies do not take place. As we increase the electron filling, doping electrons begin to occupy the moiré lattice sites, and we notice that X1 emission eventually ceases after a certain threshold gate voltage in agreement with what we expect from the first described scenario.

Under these considerations, the physical situation presented in Fig. 2a, does not correspond to a hybrid Mott state, but a purely fermionic Mott insulator probed through a highly diluted exciton population. Therefore, the cessation of X1 PL occurs because, at $\nu_e=1$, there are no available empty moire sites to create new excitons that could produce this PL signal. To generate an exciton in this condition, the system has to overcome the strong on-site Coulomb repulsion between an electron and an exciton, resulting in the displayed energy shift. In this context, X2 represents an exciton coexisting with an electron in the same moire site. To present a more clear interpretation of this figure to the reader, we are modifying the paragraph of the RESULTS section in the main text to include:

“Comparing Fig. 2a and Fig. 2c, one can observe that in the former case, the emergence of the X_2 peak corresponds to a sharp suppression of X_1 , while in the latter case, both peaks coexist. This observation gives information on the nature of double occupancy: in the first scenario, an exciton forms in the presence of an electron, and after its recombination, there are no other optical excitations in the system. In contrast, the coexistence of both peaks in panel c indicates that upon double exciton occupancy, the recombination of X_2 precedes the recombination of X_1 ”

The additional feature observed near $v_e=1$, is consistent with PL emission from midgap states (localized excitons). On one hand, this feature is only discernible under very low-power conditions, and on the other hand, its features drastically change depending on the spot of the sample. To provide a deeper interpretation of this features, we added the following discussion to the first paragraph of the RESULTS section:

“The dim mid-gap features between X1 and X2 at $v_e \sim 1$ are strongly position dependent and disappear at higher power. This indicates that such emission is from localized excitons.”

4) The PL analysis is troublesome. The authors claim “assuming that the radiative decay rate of both lines is similar.” This is a bold claim. What evidence do the authors have to support that? Did the authors measure the lifetime? The higher energy PL often has a shorter lifetime and probably a lower quantum yield.

We appreciate the reviewer’s observation and their concerns regarding the assumption about the radiative decay rates of both PL lines. Indeed, the submitted manuscript contains this assumption, which is not entirely justified. We acknowledge this and took steps to rectify it.

For clarity, the only part of the work where the decay rate of the excitons becomes relevant for the analysis is the deduction of the adequate functional form for the saturation behavior of the X1 line with PL power (Fig. 3e-h). Importantly, this analysis is not performed on the X2 line. Its behavior differs from X1, so we specifically engage Γ_1 (the decay rate of X1), without reference to Γ_2 . Recognizing potential ambiguities from the prior text, we chose to delete that statement from the main document. We modified that discussion (page 4) as follows:

“The gate voltage at which the crossing takes place (V_g^{cr}) is highlighted on each panel by a vertical dashed line. This line indicates a constant ratio between the X1 and X2 populations.”

We also deleted the following statement from the document because it also relies on the mentioned assumption:

“We choose this quantity, rather than the input pump intensity because the latter is not proportional to the total number of created excitons (see Supplementary Material for further details).”

Following the reviewer’s suggestion to enrich the analysis of our results, we performed lifetime measurements on our device (Fig. R2). However, it’s important to note that these measurements cannot be directly associated with the radiative decay rate. The exciton density depends on the total decay rate, which encompasses both radiative and non-radiative components. As a result, we cannot directly discern the radiative decay rate from the lifetime measurements.

Figure R2: interlayer exciton's lifetime as a function of the gate voltage. Panel a contains the normalized time dependent intensity for each gate voltage, and panel b shows the extracted lifetime from an exponential decay fitting.

This data confirms the expected behavior suggested by the reviewer: the lifetime of X1 differs from the lifetime of X2. For example, at 0V, where all the population corresponds to single occupancies, the lifetime is around 15 ns. On the other end, at $V_g > 3$ V, all the signal comes from double occupancies, and the lifetime decreases to around 2 ns.

5) The authors also claim, "We notice that as electrons gradually fill the system (upon increasing V_g), the number of accessible single-occupancy states decreases. As a consequence, the integrated intensity of X1 reduces with increasing v_e ." The authors seem to picture the added electron occupying each site sequentially. If so, this picture is wrong. The electron is, in general, delocalized before the phase transition happens. The decrease in PL can be simply attributed to the increased nonradiative channels to the free electrons.

The presented analysis is not intended to suggest that electrons fill the system sequentially. We understand the reviewer's concern; indeed, the word "gradually" may have been misleading in this context, and we opted to remove it to avoid any misconceptions.

It is important to note that even if the electrons are itinerant, the number of available states is an important quantity in our phase space-filling model, as the repulsion between electrons and excitons still applies. Counting the electrons within a momentum or position basis does not affect our overall interpretation. The key concept is the reduction in the number of available states for low-energy excitons (X1) due to electron doping.

The validity of the interpretation of interaction-induced band formation is subject to the condition of an energy gap larger than the exciton transition linewidth and the particles' kinetic energy, something supported by the experimental data. This interpretation is further supported by the intensity data (Fig. 2e-g), which shows a monotonic decrease of I1 with increasing gate voltage.

Taking the reviewer's feedback into account, we incorporated this clarification in the revised manuscript. We modified the misleading sentence as follows:

"We notice that as electrons fill the system's phase space (upon increasing V_g), the number of accessible single-occupancy states decreases. As a consequence, the integrated intensity of X_1 reduces with increasing V_g ."

6) There are two peaks in X_2 in Fig. 3c that the authors did not discuss.

The PL peaks displayed in Fig. 3c at an energy between X_1 and X_2 , correspond to the observation described in point 3: PL emission from midgap states (localized excitons). It can be noticed how they are quickly overcome by the X_1 and X_2 emissions upon increasing PL power. We thank the referee for pointing this out, as this feature could potentially mislead the reader. To avoid this, we include the following clarifying sentence in the discussion of the figure:

"In the low power region, panel c shows the PL emission from mid-gap states also observed in Fig. 2a."

7) The diffusion pattern in Fig. 4a is weird, with PL intensity fluctuating across y . Is that spatial inhomogeneity? How could diffusion length be extracted with such a pattern?

As the reviewer points out, this irregular diffusion pattern comes from spatial inhomogeneity. Once injected under the laser spot, the excitons diffuse on both sides: positive and negative y . However, irregularities in the sample affect their diffusion, making it asymmetric. For this reason, we extract the diffusion length from the negative y data, as we mention in the main text ("Fig. 4c shows the extracted L_{X_1} as a function of V_g and for different pump intensities from the exponentially decaying spatial diffusion pattern in the negative y region").

Although we use the upper part of the diffusion pattern for the extraction of the diffusion length (using a fitting routine), the lower part shows a more evident suppression of the exciton diffusion upon increasing power. Considering such clear benchmark of the exciton incompressibility in the raw data, we decided to include it in the main text, as it provides clear experimental evidence of the effect. Despite the impossibility of fitting the data on this side of the diffusion pattern, the inhomogeneity helps to visually evidence the incompressibility (Fig. 4a-b). This qualitative observation is quantitatively corroborated afterward with the analysis presented in Fig. 4c.

We further discuss this matter in the first point of the response to Reviewer #3.

8) The authors claim that "an incompressible bosonic state is expected to have a lower diffusion length than a weakly interacting one." Shouldn't the diffusion constant be more relevant to probe the compressibility?

The reviewer brings up an important point regarding the relationship between diffusion length and compressibility. The discussion in our paper draws from the idea that an incompressible

bosonic state, characterized by its strong correlations, would inherently manifest a reduced diffusion length with respect to a system with weak interactions. This is premised on the understanding that strong correlations can impede particle mobility, thereby influencing the diffusion length.

Although we agree with the reviewer about the pertinence of the diffusion constant (or diffusion coefficient $D_x = L_x^2/\tau$) for a comprehensive understanding of the exciton mobility properties, the only situation where our interpretation of the results would be invalid is upon an eventual dependence of the lifetime τ with the particle density. In this case, the “apparent” change in the diffusion length L_x would be associated with changes in the lifetime rather than in the exciton effective group velocity. This scenario was discarded considering previous results reported in the literature. In particular, Fig. 2 of Jauregui et al., (Science 366, 870-875 (2019)) reports negligible changes in the slow decay time (more relevant to our experiment) of the interlayer excitons in moiré heterobilayers over three orders of magnitude of pumping power.

We thank the reviewer for making us notice the importance of including this discussion in our manuscript. We are adding the following clarifying sentence in the paragraph corresponding to Fig. 4:

“The choice of L_{x1} as an appropriate quantity to benchmark the incompressibility of bosonic Mott insulating states, assumes a constant exciton lifetime with varying population. This is supported by previous reports in the literature that show the independence of this quantity over three orders of magnitude of pumping power [Science 366, 870-875 (2019)].”

9) The authors claim diffusion is suppressed at higher excitation power in Fig. 4b. But this suppression could be just due to the shortened lifetime at high excitation power. That is not necessarily a sign of incompressibility. Have the authors measured the lifetime for different excitation power?

Although we performed doping-dependent lifetime measurements in our sample (Fig. R2), the requirement of a pulsed excitation makes it cumbersome to compare the power dependence of this quantity with the one of the diffusion length (collected under CW excitation). Moreover, the varying particle density experienced by the excitons along their diffusion path (in a sub-wavelength range), makes it difficult to characterize an eventual population-dependent lifetime. However, following up on the previous point, the reported independence of the exciton decay time over three orders of magnitude of pumping power, makes the diffusion length a reliable quantity for the demonstration of the system’s incompressibility. This feature implies that the modification of the diffusion pattern depends exclusively on the propagation properties of the interlayer excitons rather than their power-independent lifetime.

In the previous point, we discussed the addition of a clarifying sentence about this matter in the discussion of Fig. 4 (quoted text in point 8). We think this sentence will also contribute to the clarification of the present concern raised by the reviewer.

10) I cannot tell what the trend at $\nu=0.6$ the author talked about is. Could the author elaborate on that?

The trend we highlight in Fig. 4c regards the diffusion length for different pump powers. For low electronic occupation, the exciton diffusion length (L_x) increases with increasing power, as expected for a weakly interacting gas of bosons. As the electronic occupancy increases, the trend inverts, and increasing the power entails a reduction of L_x , a manifestation of the incompressible nature of such states. This is the trend we are trying to remark on. For the collected set of powers, this behavior becomes evident at $\nu > \sim 0.6$, although this point is only a reference to the eye.

The reviewer's question makes us realize that this discussion needs clarification in the manuscript, for which we are modifying Fig. 4c to bring the attention of the reader to this point. The new version (Fig. R4) has guides for the eye (black arrows) that evidence the mentioned behavior and an inset in the range where the power dependence inverts. Additionally, attending to the second point raised by reviewer #3, we include the crossing gate voltage V_g^{Cr} in a version of the figure added to the Supplementary Material. They are visual markers of where the incompressible states are expected to form.

Figure R3: Exciton diffusion length as a function of the gate voltage for different input intensities. The black arrows were included as guides for the eye to evidence the change in the behavior upon the formation of incompressible states. The dashed lines mark the crossing gate voltage for each power. They indicate where the incompressibility is expected to manifest.

The main text has also been modified to improve the interpretation provided to the reader. We decided to remove the misleading reference to the specific filling factor 0.6, as it does not correspond to any particular state. The discussion now reads:

“For low electronic density, the exciton diffusion length increases as the power is augmented. This trend, highlighted by an upward arrow, is in agreement with the expected behavior for weakly interacting bosons [26,27]. Remarkably, as the electronic filling factor increases, the trend completely inverts (inset). This is a direct signature of the bosonic Mott insulator formation, since the bulk is incompressible and the melting only occurs at the edge.”

11) An important question about diffusion fitting: How do the authors know the linear rate equation is suitable across more than two orders of magnitude increase of excitation power? Have the authors measured the excitation power dependence of the PL? Is that a linear function?

We appreciate the referee mentioning a possible connection between the rate equations (eq. S1) and our diffusion data; nevertheless, we do not utilize this theory for such a purpose (the rate equation is position-independent). This theoretical formalism is intended to justify the functional form of the saturation behavior of the X1 PL power upon increasing exciton density.

However, the reviewer brings up an important point that needs clarification. Indeed, the total PL power does not depend linearly on the pump intensity. We performed this characterization and presented it in the original Supplementary Material (Fig. S4). An implication of this is that the pump power is not directly proportional to the injected exciton density, i.e., the two orders of magnitude of difference in the pump power, do not imply the same difference in the exciton density. This is the reason why Fig. 3 shows different measured quantities as a function of the total PL power instead of the pump laser intensity. This last quantity is a much better indication of the excitonic density in the system since it is proportional to $\Gamma_1 n_1 + \Gamma_2 n_2$, where $\Gamma_{1(2)}$ is the decay constant of the excitons in X1(2) and $n_{1(2)}$ the respective populations. Regarding the diffusion, since Fig. S4 shows a monotonic increase in PL power with pump intensity, the key observation from Fig. 4c is still valid: at high electronic occupation, the injection of an increasing population of excitons drives the system into a Mott insulating state.

12) There is no discussion of the entire Fig. 5 in the main text.

We are adding the following discussion paragraph of Fig.5 into the main text:

“The implemented fitting algorithm allows us to track the changes in the energy of both species of excitons and the energy gap between them. These results are presented in Fig. 5. Panels a and b show the central energies of the peaks X_1 and X_2 in the space of parameters for which each peak is detectable. In the range where both of them can be detected, their energy difference ΔE (panel c) provides important information about the nature of the interactions taking place in the system. In the case of low electronic occupancy and high exciton density (top left

corner of the panel), ΔE corresponds to the exciton-exciton interaction gap ($U_{\text{ex-ex}} \sim 32$ meV). Conversely, at high ν_e and low exciton density (bottom right corner), this gap depends on the exciton-electron interaction ($U_{\text{ex-e}} \sim 29$ meV). The gradual change in the nature of the interactions taking place in the system along the parameters space is reflected in the change of ΔE . Interestingly, the largest energy gap takes place for states with high occupation of bosons and fermions (top right corner), which is consistent with a blueshift of the X_2 PL peak due to the high population of excitons with large Bohr radius repelling through dipolar interaction.”

13) What is the twist angle between WSe2 and WS2? Did the author reproduce their results in other samples?

To address the reviewer’s point in the best possible way, we performed Second Harmonic Generation (SHG) measurements on our device in order to determine the stacking angle of the component monolayers. Using a 100 fs excitation at 850 nm with variable linear polarization, we are able to reconstruct the expected behavior of the SHG angular dependence for a crystalline structure with inversion symmetry breaking (Fig. R5a).

We first identify spots on the sample with bilayer and monolayer regions by using the PL emission for the characterization (panel b). After identifying each spot, we proceed to measure the SHG spectrum while keeping the fs laser power constant at $\sim 150 \mu W$. As demonstrated in Wang et. al., *Optical Materials Express*, 9, 3, (2019) and the Supplementary Material of Jin et.al., *Nature* 567, 76–80 (2019), the stacking angle generates a constructive (destructive) interference in the SHG spectrum for 0° (60°) stacking. The strong suppression in the SHG efficiency observed in panel c, allows us to conclude that our sample corresponds to a bilayer with a 60° stacking angle (AB or H stacking).

Figure R4: SHG sample characterization for the determination of the stacking angle. a) SHG angular dependence from a representative spot in the sample, characterized by the PL spectrum (inset). b) Characteristic PL emission spectra of spots on the sample with WSe₂ monolayer (blue) or WSe₂/WS₂ bilayer (red). c) SHG corresponding to the two spots of panel b. The strong suppression of the SHG emission efficiency indicates that the sample is stacked at an angle of 60°.

We finally point out that our results are consistent with other reports from the literature. For example, the PL dependence corresponding to an H-stack sample presented in Fig. 1 of Lian et. al., arXiv:2308.10799v2, is in good agreement with the corresponding characterization in our sample, presented in Fig. S3. We consider this data very relevant to the reader, as it provides important information about the sample characteristics, and hence we are including this figure and its discussion in the Supplementary Material.

We conclude by thanking the reviewer again for these comments that substantially improved the quality of our manuscript. We answered each point and we think the new upgraded version of this work satisfies the reviewer's requirements and hence it is suitable for publication in Nature Communications.

Reviewer #2 (Remarks to the Author):

The manuscript report experimental measurements of the WS₂/WSe₂ heterobilayer with changing the electron density and exciton density (through the intensity of optical pump). The

major discovery is a 'Mott insulating' state when the total density of electron and exciton is 1.

The measurement of the exciton density seems to be indirect, because there is no clear quantitative relationship between the optical power and exciton density. Also the major evidence of the "Mott gap" is through the change of the energy in PL spectrum, without direct transport or charge compressibility measurement. But the interpretation looks promising to me, and I think the paper is suitable for natural communication.

I have a few comments/suggestions which I hope the authors could consider:

We are glad that the reviewer finds our interpretation promising and our work suitable for publication in Nature Communications. We are grateful for the provided comments and suggestions, which helped to significantly improve the quality of our manuscript. We individually addressed each point raised by the reviewer as follows:

(1) The definition of the filling ν_e is not very clear. I need to guess that it means the electron density in the conduction band, but from WS₂ or WSe₂?

In the type II band alignment of our heterostructure, the conduction band of the WS₂ has a lower energy than the one of the WSe₂ (Fig. 2 of Xu et.al., Phys.Chem.Chem.Phys., 2018, 20, 30351 (2018)). This stacking implies that upon electron doping, the charge will reside in the WS₂ (as illustrated in Fig. R5) but it will be subject to the moire potential of the bilayer structure. The filling ν_e then corresponds to the ratio between the density of electrons in the WS₂ monolayer, and the density of moire sites in the WS₂/WSe₂ heterostructure.

To make this point more clear in the manuscript, we are modifying the discussion of Fig. 1b as follows:

“Due to the type-II band alignment of the heterostructure (Fig. 1b), negative doping results in a population of electrons in the WS₂ subject to the moire potential of the bilayer. The ratio between the density of this population and the density of moire sites in the structure, determines the so-called electronic filling factor (ν_e). The optical pump results in the formation of an energetically favorable inter-layer exciton (X) [20], by pairing an electron in WS₂ and a hole in WSe₂ (represented in Fig. 1b).”

Figure R5: type II band alignment of the WS₂/WSe₂ heterobilayer. Since the bottom of the conduction band of the WS₂ has lower energy than the one of the WSe₂, the electronic population (dashed line indicates the Fermi level) will reside in the WS₂ layer, but it will be subject to the moire potential.

(2) I'm not sure the word 'bosonic Mott insulator' is a good terminology. In my understanding, the phase refers to an "insulator" with the total number of electrons and exciton is 1 per moiré site. So it is apparently not purely 'bosonic'.

We thank the reviewer for this insightful comment. As correctly pointed out, the term "bosonic Mott insulator" is strictly correct only when $\nu_e=0$, the situation in which each moiré lattice site is occupied by an exciton and no fermions populate the lattice. On the other hand, at very low excitation power and $\nu_e=1$, the system is more fittingly described as an exciton in presence of a "fermionic Mott insulator".

The terminology used in the manuscript takes into consideration that the experimental data originates exclusively from the excitonic recombination. In this case, we preferred to avoid claims which cannot be verified from the data. Since the experimental methods are limited to test the incompressibility of the excitons, we preferred to keep this term, although we are aware of the fact that the complete state involves the incompressibility of both fermions and bosons.

A simultaneous test of the incompressibility of both component particle species would require transport measurements, as the reviewer mentioned in their initial comment. Such a demonstration would be a major breakthrough, and it is one of the perspectives of our research, but lies beyond the scope of this work.

We finally thank the reviewer for the constructive comments that improved the quality of our work. We are confident that this upgraded version of our manuscript will contribute to the comprehension of Hubbard physics in these systems and will positively impact the field.

Reviewer #3 (Remarks to the Author):

"Excitonic Mott insulator in a Bose-Fermi-Hubbard system of moiré WS₂/WSe₂ heterobilayer" authored by Gao, etc. studies the correlation between excitonic states and free carriers in TMDC heterostructure, demonstrated exciton Mott insulator states, which is a hot topic. The work agrees with previously reported results and provides new insight through the spatially resolved measurements. I would support publishing the manuscript with some edits. Also, there are some questions the authors may try to address in the revision.

We thank the reviewer for this insightful examination of our work and we greatly appreciate the positive assessment and recognition of it. We are fully committed to making the necessary edits and addressing the raised questions. This feedback is crucial to the refinement of our study.

1. Fig 4 shows the spatially resolved measurement. In fig 4a and 4b, the spatial distribution seems to exhibit some features (horizontal stripes). Are those features real or they are artifacts from the measurements? If it is real, could the authors comment on the feature? Also, as the

excitons diffuse away from the pumping region, the density will decrease dramatically and may have different diffusion constants. I am curious whether it will impact the analysis of the diffusion length. Also, as the exciton density decreases, shall we expect a transition from the X2 exciton to the X1 as moving away from the pumping region?

The horizontal stripes on the lower part of the diffusion image (Fig. 4a-b) are real. We attribute this discontinuous propagation pattern on the bottom side to inhomogeneities in the sample that locally change the optical properties of the PL emission. Despite this complex pattern, the lower part of the diffusion image shows an evident suppression of the exciton propagation, which provides clear evidence of the effect we are trying to highlight. However, for the quantitative analysis of the diffusion, it is necessary to use a fitting routine, for which the smooth pattern on top is more reliable. As shown in Fig. 4c, the exponential fitting of this side (top side, i.e. $y < 0$) provides a reliable quantification of the changes in the diffusion length. To make this point more clear to the reader, we are including the following comment to the discussion of Fig. 4:

“The downward diffusion image has patterns that originate in the inhomogeneous surface of the bilayer. Although the inhomogeneities on that side hinder the extraction of L_{X1} , the optically-induced suppression of the diffusion length for constant v_e can be clearly observed in this region (Fig. 4a-b) ... For the quantitative analysis of the diffusion, it is necessary to use a fitting routine, for which the smooth pattern on top of the injection point ($y < 0$) is more reliable. Figure 4c shows the extracted L_{X1} as a function of V_g for different pump intensities from the exponentially decaying spatial diffusion pattern in this region. We provide more details about the analysis of the diffusion data in Section IX of the Supplementary Material.”

The reviewer’s concern about the change of the lifetime and diffusion constant with propagation is valid. Unfortunately, the large exciton mass hinders the possibility of studying independently different spots along the diffusion, as they have sub-diffraction limit features. However, previous works have shown that the lifetime of interlayer excitons in moire TMD heterostructures, has a weak dependence on the population of excitons injected in the system (Jauregui et al., Science 366, 870-875 (2019)). This observation allows us to conclude that the lifetime does not have a pump-intensity dependence and that the changes in the diffusion length come mainly from changes in the exciton mobility.

We also appreciate the insightful feedback on the X2 to X1 population transfer. Based on this concern, we analyzed the data with an emphasis on possible energy transitions of excitons along the image of the diffusion. By looking at the data (Fig. R6), it becomes evident that as the excitonic population diffuses, part of the X2 particles decay into the X1 state during the diffusion. The panels b and d show the integrated intensity of the diffusion pattern in the spectral regions X1 (red) and X2 (blue) for $v_e=0$ and different pump intensities. We can observe how the increase in the pump intensity generates changes in the X1 diffusion. The population outside the pump spot becomes larger than under it, a clear indication of population transfer from X2 to X1 along the diffusion pattern.

Figure R6: Diffusion of the system for $v_e=0$ and two different pumping intensities. a) $9.42 \mu\text{W}/\mu\text{m}^2$ and c) $1270 \mu\text{W}/\mu\text{m}^2$. The right panels (b and d) show the horizontally integrated intensity. The transfer of population from X2 to X1 becomes evident in panel d, where the X1 intensity is larger outside the pumping spot due to a large reservoir of X2 excitons that decay into X1 as they propagate.

The verification of this behavior, however, does not change the interpretation of our results. The effect of the population transfer from X2 to X1 would favor an apparently longer propagation constant for X1. The X2 population would act as a reservoir for X1, which would recover part of its population as it diffuses, entailing an apparent longer diffusion. Therefore, this observation is an indication of the robustness of the demonstrated effect, for which we are glad that the reviewer brought it up. Considering the importance of this observation, we decided to include it in the work. We are adding Fig. R6 to the Supplementary Material with the following discussion:

“Delving deeper into the spatial diffusion, we identify a transfer of population from X2 to X1 as the excitons move away from the pump region. As observed in Fig. S10, for high pump power, the population of X1 away from the excitation spot is larger than under it, a manifestation of the mentioned effect. At the excitation spot, the high exciton density leads to the formation of a localized Mott insulating state and therefore a large population of X₂. However, as the distance from the pumping region increases, part of the population transfers from X₂ to X₁. Notice how, for high excitation intensity, the X₁ signal becomes stronger away from the injection spot than under it. In this case, the X₂ population is acting as a reservoir for X₁ along the diffusion path. Importantly, this behavior does not change the interpretation of our results, but constitutes proof

of the robustness of the effect; the population transfer from X_2 to X_1 would favor an apparently longer propagation constant for X_1 , which, at most, would hinder the measured suppression of the diffusion.”

We thank the reviewer again for this interesting observation that enriches the content of our manuscript and brings new perspectives into it.

2. In Fig 4c, the variation of the X_1 diffusion length seems to have a similar pattern. After the electron doping reaches ~ 0.1 filling, the diffusion lengths decrease and then hit a plateau. When electron filling reaches ~ 0.3 it starts to increase again. Is there any explanation for this phenomenon? Could it be possible to have some intermedia state at the plateau region, for example, correlated electron states freeze the exciton? Also, the diffusion lengths vs doping curves are very similar in the low doping region except for the overall increase of diffusion length with the increasing exciton injection. The overall increase of diffusion length seems to suggest ex-ex interaction is strong enough to change the diffusion length. Meanwhile, the doping-induced X_1 to X_2 transition seems ex-e interaction significantly changes the available site for single occupied excitons. One may expect the transition features of diffusion length will shift to lower electron doping levels at higher pumping densities. The author may add the X_1 to X_2 transition voltages for the different pumping power in figure 4c to help readers to interpret the transitions.

The reviewer brings up an interesting observation. Indeed, the X_1 diffusion length has non-trivial features even at low electron density. Although we agree that this could be related to fractional correlated states in the system, our current data set does not provide conclusive evidence on this regard, reason for which we refrained from including this hypothesis into the manuscript. This is one of the main perspectives that we have for our near-future research projects. Another intriguing feature is the behavior of the gate-dependent PL spectrum (Fig. 2 and Fig. S3), for which there are modulations at fractional filling factors consistent with the reviewer’s interpretation.

Also, as pointed out by the reviewer, the longer diffusion length achieved through increasing pump intensity at low electron filing, is a manifestation of the exciton-exciton interaction (U_{ex-ex}). As expected for a weakly interacting bosonic population, increasing density entails longer diffusion constant (Jauregui et.al., Science 366, 870 (2019), Unuchek et.al., Nat. Nanotec., 14, 1104–1109 (2019)). To emphasize this observation, we are including an additional remark to the discussion of Fig. 4.

“For low electronic density, the exciton diffusion length increases as the power is augmented. This trend, highlighted by the upward arrow, is in agreement with the expected behavior for weakly interacting bosons [26, 27]. Remarkably, as the electronic filling factor increases, the trend completely inverts (inset).”

Finally, attending suggestions of reviewers #1 and #3, we made modifications in Fig. 4 to highlight the suppression of the exciton diffusion. In particular we are including an inset in the region of interest and guides for the eye that evidence the effect. We are including a version of

Fig. 4c in the Supplementary Material with indicators of V_g^{cr} : the transition gate voltage at each pump power. These guides for the eye provide a visual mark of where the incompressible states are expected to form. Following also the suggestion of Reviewer #1, we added arrows to indicate the opposite behavior of the diffusion length upon increasing electron density. The new version of this panel can be observed in Fig. R3.

3. In Fig 2 as the doping increases the X1 excitons exhibit an energy redshift. Such phenomena are more obvious in the high pumping power region. Is there an explanation for the shift? Is the shift from electron-exciton interaction or the interaction of the exciton dipole with the applied gate voltage?

For low electronic filling ($\nu_e \sim 0$) and high pump intensity, the X_1 line undergoes a blueshift due to mean field interaction. More specifically, $U_{\text{ex-ex}} \langle \hat{x}^\dagger \hat{x} \rangle$. As we increase the electronic doping, fewer sites are available to create X_1 excitons and on those occupied sites, only X_2 is created. Consequently, the effective population of X_1 excitons is decreased and therefore the meanfield shift is suppressed to the point that at high filling ($\nu_e \sim 1$) the X_1 energy is the same as the low power X_1 energy.

We emphasize that a fully quantum description of this system is lacking. Such theoretical model must consider all the following components:

- The non-negligible tunneling of electrons and excitons between sites.
- The different forms of X_2 , one that is due to an electronic occupation ($U_{\text{ex-e}}$) and the other that is due to exciton-exciton interaction ($U_{\text{ex-ex}}$).
- Phase space filling of each site: Our preliminary theoretical estimation indicates that since the Bohr radius is comparable to Wannier orbital size, the bosonic description of such excitons is not accurate.

Based on this point, we included the following discussion in the first paragraph of the RESULTS section:

“At $\nu_e=0$, the X_1 peak in Fig. 2c is blueshifted with respect to Fig. 2a. We associate this feature with a mean-field effect due to exciton-exciton interaction. As we increase the electronic doping, fewer sites are available to create X_1 excitons and on those occupied sites, only X_2 is created. Consequently, the effective population of X_1 excitons is decreased and therefore the meanfield shift is suppressed to the point that at high filling ($\nu_e \sim 1$) the X_1 energy is the same as in the case of low pump intensity.”

We thank the reviewer for bringing this point to our attention. The implemented modification will clarify this point to the reader.

4. It is inevitable to compare the result of the manuscript to ref 36, which studies similar physics in the same material system. It seems that the pumping density(exciton injection rate) is quite different between the manuscript and ref. 36(CW laser pump). Could the authors calibrate exciton density to compare the results? Also, did the author lock the CCD to the pumping pulse? If not, will the dynamics of the excitons impact the result? For example, after the pump pulse, as the excitons recombine and diffuse away from the pumping region, the overall exciton density will decrease. If start within the X2 region, would it lead to a transition from X2 to X1? If the CCD keeps integrating PL signal, and such transition exists, would it be responsible for the long tails of X1 and X2(for example fig 2c, fig. 3b,c)?

As the reviewer indicates, the mentioned work studies a similar system under a pump-probe experimental configuration. In that case, the authors use a CW power-variable pump spot and a low power CW probe to obtain information about the state of the system. This configuration can be interpreted as a differential measurement. To make a proper comparison in our system, and properly address the reviewer's question, we collected a set of data in CW excitation regime. The results are displayed in Fig. R7. For increasing power, the measured differential PL spectrum $(PL(power_i) - PL(power_{i-1}))/\Delta power$, shows consistency of our results with the ones from ref. 36. Comparing the left column of Fig. R7 with Fig. 2A of the reference, we can conclude that the obtained behavior is the same in both experiments.

The reviewer's observation regarding the pump intensities is very relevant. There are multiple factors that can affect the effective laser intensity on the sample. To mention some of them, the thickness of the hBN, the presence or absence of graphene contacts under the spot, the pump energy and even the set of optics affect the incident power on the bilayer, for which, from our point of view, the dissimilitude of the power calibration does not represent a fundamental problem. An accurate calibration of the exciton population in any semiconductor system involves different experimental challenges. The most important one is the impossibility to experimentally isolate the radiative from the non-radiative losses. For this reason, such a calibration lies beyond of the scope of this work.

Regarding the second point, unfortunately, we are limited by the CCD acquisition speed. For example, as shown in Fig. R2, the longest measured lifetime of the interlayer excitons is ~15 ns, a speed that no CCD can achieve. Reference 36 uses a low-frequency modulation of the probe power and they detect in synchronization with this modulation. It is not clear for us how this represents an advantage, since the times of illumination are several order of magnitude longer than the typical times in the system's dynamics.

As the reviewer correctly pointed out, the population can transfer from X2 to X1 as it diffuses, and this definitely limits the precision with which we measure L_x . However, due to the aforementioned considerations, this effect can only entail an apparent increase in the value of L_x . The presented experimental method is robust enough to this effect, because in spite of it, we evidence a decrease on this value.

Figure R7: power dependent PL map. The left column shows the raw data for $\nu = 0$ (a), $\nu = 0.7$ (c), $\nu = 1$ (e), while the right column (b,d and f) displays the differential power-dependent PL, an analog method to the one presented in ref. 36. The results are consistent with the mentioned reference.

5. In ref 36, they divide the phase diagram into 3 regions and separated the exciton-exicton interaction from the exciton-electron interaction. The authors claimed, “within our experimental resolution, these latter two situations are not spectrally resolvable.” However, fig.5c indicates that the ex-ex and ex-e states can be separated by comparing X1 X2 energy as well as the filling numbers of exciton and electron. Could the authors clarify their argument? The definition of ex-ex interaction may need to be clarified as well. When authors discuss the blueshift of X1 exciton energy in pumping power dependence measurement(fig3), $U_{\text{ex-ex}}$ is quoted. I would assume this is the interaction between excitons at different sites. However, in fig 5c, the $U_{\text{ex-ex}}$ seems to be ex-ex interaction of exciton from the same site. Also, there is no discussion about fig. 5. in the manuscript.

The reviewer raises an important point that needs clarification in the manuscript. We agree that it can be misleading for the reader that we claim we cannot resolve the PL from ex-ex and ex-e double occupancies, but then we give different values for these interaction energies in Fig. 5. We will elaborate on this:

That statement is intended to remark on the fact that we never detect two resolvable peaks for X2 in a single spectrum. This is due to the broad linewidth of this emission with respect to the energy difference between $U_{\text{ex-e}}$ and $U_{\text{ex-ex}}$. The PL can then be interpreted as the averaged contribution of these two emissions, with weights determined by the combination of Vg and pump intensity. The effective central energy of the peak will then depend on this.

In the two limiting cases (1: low pump intensity and $v \sim 1$, and 2: high pump intensity and $v \sim 0$), it is correct to assume that there is only one interaction present in the system ($U_{\text{ex-e}}$ and $U_{\text{ex-ex}}$, respectively). Thanks to this, and in spite of the broad emission line, we can assign accurate values to both interaction strengths in these two regimes. To clarify this in the manuscript, we are changing the referred statement as follows:

“In this case, the PL emission corresponds to mixed contributions from exciton-exciton and exciton-electron interaction ($U_{\text{ex-ex}}$ and $U_{\text{ex-e}}$); the individual peaks cannot be distinguished in a single spectrum due to the broadness of linewidths. Therefore.....”

We now turn our attention to the definition of $U_{\text{ex-ex}}$. This is a fundamental point because it is related to the physical model that we use to interpret our results: the Hubbard model. As the reviewer points out, we attribute both the gradual blueshift of the PL upon increasing excitonic population, and the existence of a gapped states, to the exciton-exciton interaction $U_{\text{ex-ex}}$. Importantly, this does not represent any contradiction. In the low occupation regime, the on-site repulsive interaction entails a linear blueshift due to the delocalized nature of the state. Upon high occupation, the interaction leads to a gap in the energy, i.e., the formation of an upper Hubbard band.

We will better address this discussion by including the missing remarks on Fig. 5. We added a new paragraph to the manuscript based on the comments of the three reviewers. The exact text is included in the point 12 of the response to the Reviewer #1.

We are also adding a more comprehensive theoretical discussion to address this point in the Supplementary Material of the new version of the manuscript.

In the absence of doping electrons, the PL spectrum displays a particular behavior for increasing excitonic population (Fig. 3a of main text): a blueshift at low occupancy and the emergence of a PL peak from a gapped state after a threshold PL power. In this “reduced” system, where the population is fully bosonic, we can model the system and account for the blueshift and the spectral gap by using the following Lindbladian master equation:

$$\partial_t \hat{\rho} = -i[\hat{H}, \hat{\rho}] + \sum_n \hat{\mathcal{L}}_n[\hat{\rho}],$$

where \hat{H} corresponds to the Bose-Hubbard Hamiltonian:

$$\hat{H} = \omega_x \hat{x}^\dagger \hat{x} + \frac{U_{\text{ex-ex}}}{2} \hat{x}^\dagger \hat{x}^\dagger \hat{x} \hat{x}$$

and the operators $\hat{x}(\hat{x}^\dagger)$ stand for the annihilation (creation) of an exciton particle, ω_x is their energy and $U_{\text{ex-ex}}$ is the on-site particle repulsion interaction energy. We consider two Lindbladian terms to take into account the laser pump and exciton loss. Specifically, the operator for channel n is:

$$\hat{\mathcal{L}}_n[\hat{\rho}] = \hat{C}_n \hat{\rho} \hat{C}_n^\dagger - \frac{1}{2} \{ \hat{C}_n^\dagger \hat{C}_n, \hat{\rho} \},$$

that includes the jump operators accounting for two incoherent processes: laser pump and exciton losses, with operators \hat{C}_p and \hat{C}_l , respectively:

$$\hat{C}_p = \sqrt{\Gamma_p} \hat{x}^\dagger, \quad \hat{C}_l = \sqrt{\Gamma_l} \hat{x}.$$

Using the quantum regression theorem for the expected value $\langle \hat{x}^\dagger \hat{x} \rangle$, we calculate the spectral function and extract the central energy of the main peak. Given the set of theory data, we perform a routine for fitting the experimental results, obtaining the set:

$$U_{\text{ex-ex}} \simeq 32.4 \text{ (meV)}, \quad \Gamma_l \simeq 10.1 \text{ (meV)}, \quad \omega_x \simeq 1460 \text{ (meV)}$$

The result, displayed in Fig. R7 shows a very good agreement with the experimental results.

Figure R8: Continuous line: Main peak position of the calculated spectral function for a Bose-Hubbard model under the Master equation formalism. The blue circles correspond to the obtained experimental data (from Fig. 3) for the PL power dependence of the spectrum at $v_e=0$.

This figure provides important insights into the nature of $U_{\text{ex-ex}}$. It shows that the on-site exciton repulsion leads to a spectral blueshift even in the delocalized-excitons regime, due to the overlapping of the particles wavefunction.

Finally, we would like to emphasize that such a master equation cannot explain the intensity-dependent behavior of X1. A unified model including higher-order jump operators (and hence more fitting parameters) may be required. We anticipate this master equation description and the rate equations presented in section VIII of the Supplementary Material being simplifications of a unified theory, which needs further investigation. We are including a comment on this regard in the discussion section:

“Moreover, a quantum microscopic model capable of fully describing such a driven-dissipative Bose-Fermi mixture remains an open area of research.”

6. Since it seems to be possible to separate the ex-ex, ex-e states. Could authors comment on the X2 diffusion data in SI and whether any feature could be correlated to the two different states?

For the sake of completeness, we included the data corresponding to the diffusion of the X2 excitons in the Supplementary Material (Fig. S9). Upon close inspection, one can identify a pronounced dip around $\nu=1$ across all pumping power levels. This feature strongly suggests that in the fermionic Mott insulator regime, where the PL is predominantly influenced by $U_{\text{ex-e}}$ interactions, the charge-ordered state entails a measurable suppression of the X2 diffusion. Despite this interesting observation, the data is not enough for a comprehensive analysis of the underlying phenomena, and further experimental and theoretical investigation is required on this front.

In conclusion, the manuscript presents high-quality experimental data. The correlation between excitonic states and free carriers in the 2D TMDC heterostructure is of broad interest and is under heavy investigation. The main conclusion agrees with recent studies on the same topic. More discussion about the novel spatially resolved measurement could further improve the manuscript.

We thank the reviewer for such positive feedback on the quality of our experimental data. We agree that the correlation between excitonic states and free carriers in 2D TMDC heterostructures is a crucial and timely subject of investigation. The suggestion on emphasizing our spatially resolved measurements is welcomed. We expanded our discussion in the manuscript to highlight the significance and outlook of the presented spatially resolved diffusion measurements, setting our work apart and providing novel perspectives. We are adding the following emphasizing remark:

“The novel experimental diffusion method used to benchmark the excitonic incompressibility, opens exciting perspectives for the simulation of complex dynamics in many-body quantum systems that range from a single bosonic particle in a Fermi sea, to a strongly interacting gas of bosons.”

Once again, we appreciate such constructive comments from the reviewer. They will greatly improve the comprehensiveness and impact of our study.

On behalf of the authors

Dr. Daniel Gustavo Suárez-Forero

REVIEWER COMMENTS

Reviewer #1 (Remarks to the Author):

I appreciate the authors' efforts to address my previous questions and improve the manuscript. The current manuscript is more complete compared with the previous version, especially considering the added discussion of Figure 5 that was missing in the previous version.

However, I cannot recommend publishing this work in Nature Communications as the experimental data do not convincingly support the interpretation and conclusion. In fact, the major claim contradicts the conclusion of previous results, and the discrepancy could be explained by extrinsic effects.

The revised work now claims the U_{ex-ex} to be ~ 32 meV and U_{ex-e} to be ~ 29 meV (cannot be resolved in the spectra). The U_{ex-ex} reported here is closer to the U_{e-ex} previously reported (32-40 meV) but significantly larger than the previously reported U_{ex-ex} (14-20 meV) (Xiong et al., Science 380,860-864 (2023)). This confirms my previous suspicion that the observed high energy PL peak is likely from doping-inhomogeneity or photo-doping effect (the authors agreed, but I think they misunderstood the meaning of photo-doping).

The extrinsic effect interpretation is supported by the defects observed in the sample studied here, as the authors assigned the additional feature in PL observed. It is also consistent with the significant inhomogeneity observed in the diffusion measurements. The authors claim that the defect is consistent from different spots. How about different samples? It reads as if the data are all from one sample. If so, it cannot be justified considering the time lag between this work and the previous work.

I also have problems with the diffusion measurements. I am not sure if the authors understood my previous questions about the rate equation and the effect of nonlinearity. What is the exciton density for each of the excitation powers studied in Fig. 4c? If the PL is significantly nonlinear at the high excitation power (exciton density), it suggests the nonlinear process dominates. How did the authors extract "diffusion length" if the linear diffusion equation is not valid anymore? I am also surprised that the authors assume the same lifetime across this large range of excitation power to be the same. Why not measure the power dependence of the lifetime of this sample directly instead of citing the previous reference? The previous work was for a different material system and might be for a different range of exciton density. Also, how to justify the lifetime of the pulsed excitation to be the same as the CW diffusion measurement?

An example of the importance of the lifetime is that the crossover region in Fig. 4c overlaps with the lifetime drop in Fig. R2b. For the gate dependence, the lifetime difference also needs to be considered.

Reviewer #2 (Remarks to the Author):

The authors have responded to my questions and comments in a satisfactory way. I recommend the paper to be published in Nature communications.

Reviewer #3 (Remarks to the Author):

The authors have fully addressed all the questions I had. The edits they have made have also improved the manuscript. I would recommend that the manuscript be published in Nature Communications.

Reviewer #1 (Remarks to the Author):

I appreciate the authors' efforts to address my previous questions and improve the manuscript. The current manuscript is more complete compared with the previous version, especially considering the added discussion of Figure 5 that was missing in the previous version.

We thank the reviewer for the thorough evaluation of our manuscript and for highlighting the improvements in our work after the previous round of review. We value this new feedback and appreciate the opportunity to address it in detail.

However, I cannot recommend publishing this work in Nature Communications as the experimental data do not convincingly support the interpretation and conclusion. In fact, the major claim contradicts the conclusion of previous results, and the discrepancy could be explained by extrinsic effects. The revised work now claims the $U_{\text{ex-ex}}$ to be $U_{\text{ex-e}}$ to be ~ 29 meV (cannot be resolved in the spectra). The $U_{\text{ex-ex}}$ reported here is closer to the $U_{\text{e-ex}}$ previously reported (32-40 meV) but significantly larger than the previously reported $U_{\text{ex-ex}}$ (14-20 meV) (Xiong et al., Science 380,860-864 (2023)). This confirms my previous suspicion that the observed high energy PL peak is likely from doping-inhomogeneity or photo-doping effect (the authors agreed, but I think they misunderstood the meaning of photo-doping).

We acknowledge the discrepancy in the exciton-exciton repulsion energy measured in our device compared to the cited paper (Xiong et al., Science 380,860-864 (2023) that we will refer to as UCSB paper). However, we would like to bring to the reviewer's attention **two recent works on an identical system (WS₂/WSe₂ heterobilayer with H-stacking) that report $U_{\text{ex-ex}}$ values fully consistent with ours**. Importantly, both references were published **after** our results were submitted to Nature Communications. This point is particularly important because the reviewer is bringing up that reference as an indication of possible contradictions in our results. We include textual quotes from the aforementioned works:

Lian et.al. *Valley-polarized excitonic Mott insulator in WS₂/WSe₂ moiré superlattice*, Nature Physics, 2023: **"The IX-IX repulsion energy ($U_{\text{ex-ex}}$)**, which we estimate from the energy difference between Xi_2 and Xi_1 , is about 44 meV for the R-stacked device R1 and about **32 meV for the H-stacked device H1.**"

Park et. al. *Dipole ladders with large Hubbard interaction in a moiré exciton lattice*, Nature Physics, 19, 2023: "We denote the onsite repulsion energy as the exciton Hubbard U and the IX1 energy as EX . The energy of the two-exciton state is $E_{2X} = 2EX + U$ (Fig. 2c, inset). The recombination cascade of the two-exciton state then emits two photons at energy $EX + U$ and EX , corresponding to IX2 and IX1, respectively. From the

experiment, **we determine U to be between 30 and 37 meV at charge neutrality, depending on the exciton density....**".

As the reviewer can notice, the reported values in the literature disregard any contradiction in our results.

To include even stronger evidence, we fabricated a new device with the same stacking order (Fig. S1a) and performed photoluminescence measurements, which we are including in the Supplementary Information of the new version of the manuscript. From the data, displayed in Fig. S1b-c, we obtain a value of 34 meV for Ue-ex and 35 meV for Uex-ex. These values are in full agreement with the data reported in the previous versions of this work (device D1) and with the cited literature.

Figure S1. (a) Optical image of the newly fabricated device (D2) (b) Gate Voltage-dependent normalized PL in Device D2 for a pump intensity of $0.048 \mu\text{W}/\mu\text{m}^2$, the splitting between X1 and X2 (Ue-ex) is 34 meV. (c) Power-dependent normalized PL in Device D2 at $\nu_e = 0$. The splitting between X1 and X2 (Uex-ex) is 35 meV.

In spite of the discrepancy in the value of Uex-ex between our results and the UCSB paper, the overall results are in mutual agreement. As illustrated in Figure R7 of our previous response, we observed analogous behavior (Fig. R7b,d,f) to what was reported in Fig. 2a and Fig. 1d of the UCSB work. This point was extensively discussed in our previous response letter.

Figure S1 allows to discard the reviewer's hypothesis of doping inhomogeneities causing the formation of a gapped state, as panel b does not show any state at the transition gate voltages. If the measured value of Uex-ex were consequence of such inhomogeneity, the device D2 would present a different interaction strength; but as shown in panel c, that is not the case.

Regarding the photo-doping effect, we appreciate the reviewer's comment, but we have to point out that we did not misunderstand its meaning; we will go over this point again: if the hypothesis of photo-doping were accurate—i.e. if the number of electrons created under the same excitation were more numerous than the holes (or viceversa)—we

would expect the disappearance of X1 at 0V, in consistency with our gate voltage-dependent PL spectrum at very low power (Fig. 2a of the main text or Fig. S1b of this response). Contrary to this expectation, as evidenced in Fig. 3a, the X1 emission remains significantly stronger than X2 at 0V. This observation leads us to conclude that the blueshift we observe is indeed a result of the double occupancy of excitons. Moreover, it is important to note that we utilize an excitation pump centered at 720 nm, resonant with the WSe2 intralayer exciton. Unlike a scenario with a high-energy pump, photo-doping is not a plausible explanation in this case.

The extrinsic effect interpretation is supported by the defects observed in the sample studied here, as the authors assigned the additional feature in PL observed. It is also consistent with the significant inhomogeneity observed in the diffusion measurements. The authors claim that the defect is consistent from different spots. How about different samples? It reads as if the data are all from one sample. If so, it cannot be justified considering the time lag between this work and the previous work.

As highlighted in the last point, we have incorporated data from an additional device to prove the robustness of our findings. The observed similar behavior, consistent with the patterns illustrated in the main paper and devoid of mid-gap states, strengthens our argument that the observed features are not from defects, photo-doping, or sample inhomogeneity.

For enhanced clarity, and to address the reviewer's concern about the inhomogeneity in diffusion measurement, we present the 3D diffusion map data of both the original and new device (as indicated in Figure S2), demonstrating their consistency. Notably, at a low doping level, we observe an increase in diffusion length with escalating power, while at a high doping level, the diffusion length shows a decrease with increasing power. We consider that showing the consistency of our results in two different devices will provide a better degree of confidence to our observations, for which we are including Fig. S2 in the Supplementary Material of the most updated version of the manuscript.

Figure S2. Power and gate voltage-dependent diffusion length of X_{tot} in device D1 (same as the device in the main text) and device D2. The difference in the magnitude of the power range is because the exciton lifetime of D2 is about 30 times longer than D1.

I also have problems with the diffusion measurements. I am not sure if the authors understood my previous questions about the rate equation and the effect of nonlinearity. What is the exciton density for each of the excitation powers studied in Fig. 4c? If the PL is significantly nonlinear at the high excitation power (exciton density), it suggests the nonlinear process dominates. How did the authors extract “diffusion length” if the linear diffusion equation is not valid anymore? I am also surprised that the authors assume the same lifetime across this large range of excitation power to be the same. Why not measure the power dependence of the lifetime of this sample directly instead of citing the previous reference? The previous work was for a different material system and might be for a different range of exciton density. Also, how to justify the lifetime of the pulsed excitation to be the same as the CW diffusion measurement?

An example of the importance of the lifetime is that the crossover region in Fig. 4c overlaps with the lifetime drop in Fig. R2b. For the gate dependence, the lifetime difference also needs to be considered.

Regarding the exciton density, by comparing the blueshift energy Δ induced by the dipole-dipole interaction of excitons with U_{ex-ex} , we can extract the exciton population per moire site $\langle a^\dagger a \rangle = \frac{\Delta}{U_{ex-ex}}$ as discussed in section X of the Supplementary Material.

The measured blueshift for D1, whose power-dependent spectra for CW pump can be found in Fig. R1 of our previous response, have the following values: 3.98 meV (for $I = 56.6 \mu\text{W}/\mu\text{m}^2$), 5.65 meV (for $I = 406.4 \mu\text{W}/\mu\text{m}^2$), 6.58 meV (for $I = 674.9 \mu\text{W}/\mu\text{m}^2$), 6.97 meV (for $I = 972.5 \mu\text{W}/\mu\text{m}^2$), 7.13 meV (for $I = 1270.0 \mu\text{W}/\mu\text{m}^2$), and 8.03 meV (for $I = 1888.3 \mu\text{W}/\mu\text{m}^2$). Using the illustrated relation, we can infer the values of $\langle a^\dagger a \rangle$ for which we obtain: 0.50, 0.55, 0.58, 0.59, 0.60 and 0.63, correspondingly.

This model is widely used for the calibration of the excitonic population in these structures, but it's important to mention that this description lacks accuracy in the range of large excitonic occupation. In this regime, a full master equation formalism is necessary, and this is precisely a perspective for an immediate research subject.

To alleviate any confusion surrounding the extraction of the diffusion length, we want to clarify our methodology. We agree with the referee that we are not in the linear regime and we do not assume a linear diffusion equation that is usually employed to extract the diffusion constant. Instead, we extract the ‘diffusion length’ that is a well-defined

parameter and it is determined as the $1/e$ value extracted from the spatial photoluminescence (PL) emission intensity profile.

We emphasize again that we refrain from reporting a diffusion coefficient and instead we focus on the ‘diffusion length’ which does not require lifetime measurements. Nevertheless, we report the lifetime measurement for completeness. We conducted power-dependent lifetime measurements, as illustrated in Figure S3. The results indicate a noticeable decrease in lifetime with increasing gate voltage (electron doping). However, with increasing power, we observe no significant change in lifetime.

Figure S3. Time-dependent PL normalized at $t = 0$ for different peak powers at 0V (a), 1.46V (b), and 1.98V (c). For this measurement, we use a pulsed laser with a 500-kHz repetition rate and 100 ps “on” time. The power we quoted is the peak power. Dashed white lines correspond to double exponential fits. (d) Lifetime values at different powers and different gate voltages.

In conclusion, we have endeavored to address each of the reviewer’s concerns comprehensively. We are confident that these clarifications enhance the understanding of our work and thank the reviewer for the time and consideration.

On behalf of the authors

Dr. Daniel G. Suárez-Forero

REVIEWER COMMENTS

Reviewer #1 (Remarks to the Author):

The authors did not understand my comments and concerns. To keep the review at a reasonable length, I will only focus on the major issues below.

Overall, I cannot recommend the publication of this work in Nature Communications. The results of this work contradict the previous publications. The data analysis is often based on assumptions and is oversimplified. The data quality and analysis do not meet the standard of Nature Communications.

(1) Discrepancy of the U_{ex-ex} and U_{ex-e} . It is not the specific value of U_{ex-ex} that contradicts previous results. For different samples, it is expected that there will be variations, as shown in the Nature Physics paper by Park et. al (Dipole ladders with large Hubbard interaction in a moiré exciton lattice).

In this work, the authors obtained the same U_{ex-ex} and U_{e-ex} with the uncertainty considered, (29 meV and 29 meV for the first device, 35 meV and 34 meV for the 2nd one). There is no reason why these two should be the same.

The UCSB work (Science by Xiong et al) clearly shows that U_{e-ex} (32-40 meV) is significantly larger than U_{ex-ex} (14 meV-20 meV). The Nature Physics paper by Lian et.al. (Valley-polarized excitonic Mott insulator in WS_2/WSe_2 moiré superlattice) also shows that for H-type WS_2/WSe_2 , the U_{e-ex} (41 meV) is significantly larger than the U_{ex-ex} (32 meV). For R-type, The U_{e-ex} is 17 meV, significantly smaller than the U_{ex-ex} (44 meV).

The reason the authors observed the same U_{e-ex} and U_{ex-ex} can be explained by the photo-doping effect due to the pulse excitation. The claimed U_{ex-ex} is the actual U_{e-ex} .

(2) I think the complication is due to the pulse excitation. The authors claim that “The pulsed excitation allows us to reach high excitonic densities with low average power, something critical to avoid slow thermal effects in the system”, but it is wrong in the context of this work. PL is a slow measurement that measures the average density of exciton, and the average exciton density is smaller for the same average power of pulse excitation compared with CW excitation, considering

the absorption saturation as a result of four orders of magnitude more photon density at the peak of pulse excitation.

(3) The author claimed “Moreover, it is important to note that we utilize an excitation pump centered at 720 nm, resonant with the WSe₂ intralayer exciton. Unlike a scenario with a high-energy pump, photo-doping is not a plausible explanation in this case.” The resonant excitation at WSe₂ intralayer excitation is far larger than the interlayer exciton energy studied here.

(4) In terms of diffusion, I still believe the significant inhomogeneity shown in Fig. 4a makes the extraction of data from diffusion unreliable.

(5) The authors claim that “Instead, we extract the ‘diffusion length’ that is a well-defined parameter and it is determined as the 1/e value extracted from the spatial photoluminescence (PL) emission intensity profile.” What does this even mean? Why is diffusion length well defined as the 1/e value?

The diffusion equation is a well-defined differential equation. The diffusion length is defined as the Gaussian width in the linear region.

(6) The authors also mention that “We agree with the referee that we are not in the linear regime and we do not assume a linear diffusion equation that is usually employed to extract the diffusion constant”.

The authors studied the diffusion under various excitation powers, with magnitude increased by more than 30 times. The reviewer does not state this has to be a linear or nonlinear region. With this large difference, the authors need to examine it experimentally, and the diffusion length needs to be analyzed accordingly. There may be a linear to nonlinear crossover.

(7) TRPL data in Fig. S3 clearly show two decay times. Why is only the slow component assigned to the exciton lifetime? The fast component is different.

(8) For the extracted exciton density, the authors obtained 0.5 for excitation power density 56.6 $\mu\text{W}/\text{cm}^2$ and 0.63 for 1888.3 $\mu\text{W}/\text{cm}^2$. If the lifetime is the same as the authors claimed, what does it mean?

Response letter to reviewers

Reviewer #1 (Remarks to the Author):

The authors did not understand my comments and concerns. To keep the review at a reasonable length, I will only focus on the major issues below.

Overall, I cannot recommend the publication of this work in Nature Communications. The results of this work contradict the previous publications. The data analysis is often based on assumptions and is oversimplified. The data quality and analysis do not meet the standard of Nature Communications.

We value the reviewer's feedback. We will address each of the concerns in detail.

(1) Discrepancy of the U_{ex-ex} and U_{ex-e} . It is not the specific value of U_{ex-ex} that contradicts previous results. For different samples, it is expected that there will be variations, as shown in the Nature Physics paper by Park et. Al (Dipole ladders with large Hubbard interaction in a moiré exciton lattice).

In this work, the authors obtained the same U_{ex-ex} and U_{e-ex} with the uncertainty considered, (29 meV and 29 meV for the first device, 35 meV and 34 meV for the 2nd one). There is no reason why these two should be the same.

The UCSB work (Science by Xiong et al) clearly shows that U_{e-ex} (32-40 meV) is significantly larger than U_{ex-ex} (14 meV-20 meV). The Nature Physics paper by Lian et.al. (Valley-polarized excitonic Mott insulator in WS₂/WSe₂ moiré superlattice) also shows that for H-type WS₂/WSe₂, the U_{e-ex} (41 meV) is significantly larger than the U_{ex-ex} (32 meV). For R-type, The U_{e-ex} is 17 meV, significantly smaller than the U_{ex-ex} (44 meV).

The reason the authors observed the same U_{e-ex} and U_{ex-ex} can be explained by the photo-doping effect due to the pulse excitation. The claimed U_{ex-ex} is the actual U_{e-ex} .

The reported values for the interaction energies U_{ex-ex} and U_{ex-e} are within the range of previous experiments in the literature. We measured a value of 27 meV for U_{ex-e} and 32 meV for U_{ex-ex} (Fig. R1), i.e. a difference of ~16% between them ($U_{ex-e}/U_{ex-ex} \sim 0.84$) in device D1 (pulsed excitation). For device D2 we measured $U_{ex-e}/U_{ex-ex} \sim 1$ when pumping with a CW excitation. The Nature Physics paper by Lian *et al.* reports $U_{ex-e}/U_{ex-ex} \sim 1.28$ for H-stack and $U_{ex-e}/U_{ex-ex} \sim 0.39$ for R stack. The UCSB work by Xiong *et al.* reports $U_{ex-e}/U_{ex-ex} \sim 2.3$. We compiled these values in Table T1. As one can see, the values of U_{ex-e} and U_{ex-ex} do not have a simple trend: not only do they change with the type of stacking, but even samples with identical nominal stacking angles show large differences in their absolute values and U_{ex-e} to U_{ex-ex} ratios.

Hence, it is not possible to experimentally associate a precise value for the U_{ex-e}/U_{ex-ex} ratio in these heterostructures regardless of the excitation (pulsed or CW).

	U_{ex-e}	U_{ex-ex}	U_{ex-e}/U_{ex-ex}
Device D1	27	32	0.84
Device D2	34	35	~ 1
Park et al. (R stack)	-	30-37	-
Lian et al. (H stack)	41	32	1.28
Lian et al. (R stack)	17	44	0.39
Xiong et al. (H stack)	35	15	2.3

Table T1: Compilation of reported values of the interaction energies U_{ex-e} and U_{ex-ex} in the literature.

We regret that in the previous response, we did not emphasize that the data in device D2 was collected with CW excitation. We corrected that in the new version of the Manuscript. Despite this, the values for U_{ex-ex} and U_{ex-e} are similar. This observation allows us to conclude that the reviewer’s hypothesis of “photo-doping effect due to the pulse excitation” cannot be true. Moreover, from Table T1 it is evident that there is not a well-established value for U_{ex-ex}/U_{ex-e} .

Fig. R1 shows the normalized PL for $I = 5 \times 10^{-4} \mu W/\mu m^2$ (lowest pump power) and $\nu_e = 1$, and for $I = 93 \mu W/\mu m^2$ (lowest power for which X2 appears even at zero doping), and $\nu_e = 0$. U_{ex-e} and U_{ex-ex} have been appropriately marked in the figure.

Figure R1. Photoluminescence spectrum of device D1 under pulsed excitation showing $U_{ex-e} \sim 27\text{meV}$ and $U_{ex-ex} \sim 32\text{meV}$.

This discussion with the reviewer made us aware of the importance of emphasizing the strong variations of U_{ex-ex} and U_{ex-e} even for nominally identical samples. To make this point clear in our manuscript we are including the compilation (table T1) and Fig. R1 in Section XII of the new version of the Supplementary Material. This modification will avoid misleading conclusions from the reader based on the values of the interaction.

(2) I think the complication is due to the pulse excitation. The authors claim that “The pulsed excitation allows us to reach high excitonic densities with low average power, something critical to avoid slow thermal effects in the system”, but it is wrong in the context of this work. PL is a slow measurement that measures the average density of exciton, and the average exciton density is smaller for the same average power of pulse excitation compared with CW excitation, considering the absorption saturation as a result of four orders of magnitude more photon density at the peak of pulse excitation.

The reviewer raised this point in the first round of revision, and to properly address it, we included a comparison between data sets collected with CW and pulsed excitations. We kindly refer the reviewer to Fig. S6, where it is evident that the data is consistent in both cases. Hence, we don't have any suggestion of artifacts generated from the choice of excitation source. Additionally, as discussed in the previous point, in device D2 we used CW excitation and the results were similar.

(3) The author claimed “Moreover, it is important to note that we utilize an excitation pump centered at 720 nm, resonant with the WSe₂ intralayer exciton. Unlike a scenario with a high-energy pump, photo-doping is not a plausible explanation in this case.” The resonant excitation at WSe₂ intralayer excitation is far larger than the interlayer exciton energy studied here.

We agree with the reviewer that the intralayer exciton has larger energy than the interlayer exciton. The hole stays in the valence band of WSe₂. It is only the electron that tunnels to WS₂ and may lead to photo-doping. We addressed this issue in our previous response by pointing out that in the presence of photo-doping, we expect the reduction of X1 intensity with pump power, in consistency with the gate voltage-dependent PL intensity at low power (Fig. 2e of the main text). Contrary to this expectation, as shown in Fig.3(e-g), the X1 emission monotonically increases for the whole range of pump power; an observation completely inconsistent with a photo-doping scenario.

To further strengthen our argument, we collected data in device D2 with an **out-of-resonance pump** to induce photo-doping and compared it with the data in device D1 with a **resonant pump** (as reported in the manuscript). As can be observed in Fig. R2, the behavior of a photo-doped sample is completely different from what we observe in the reported data. We observed that X1 **decreases** at certain pump power, while X2 keeps increasing, which is an indication of photo-doping.

Figure R2. Evolution of PL intensity for X1 (red) and X2 (blue) as a function of the total excitation intensity for two different excitation regimes: a) device D2 pumped with an out-of-resonant CW laser (633nm) and b) device D1 pumped resonantly (720nm). Due to the photo-doping effect, the intensity of X1 reduces after a threshold power, in contrast to the resonant excitation, where the PL intensity increases monotonically. In both panels $\nu_e = 0.3$. Counts are divided by a factor 10^6 .

From Fig. R2, we discard any photo-doping-induced artifact in the presented experimental data. We consider that this is important information for the reader, therefore we are adding and discussing Fig. R2 in the Supplementary Material.

We note that there is another alternative raised by Referee #3 (creation of intralayer excitons instead of free carriers) that has been discussed in great detail in Fig. R4.

(4) In terms of diffusion, I still believe the significant inhomogeneity shown in Fig. 4a makes the extraction of data from diffusion unreliable.

We carefully addressed this point in the previous round of response; we included Fig. S13 to show the power and gate voltage-dependent diffusion length of X_{tot} in devices D1 and D2. Since the trend in diffusion length is reproducible over different samples and multiple spots, it cannot be an artifact arising from local inhomogeneity. We wonder why the reviewer does not refer to this observation at all. Additionally, as we have mentioned on different occasions, we extracted the diffusion length by fitting the intensity of the PL diffusion only along the negative y-axis in Fig. 4a because of good homogeneity in that region of the sample.

(5) The authors claim that "Instead, we extract the 'diffusion length' that is a well-defined parameter and it is determined as the 1/e value extracted from the spatial photoluminescence (PL) emission intensity profile." What does this even mean? Why is diffusion length well defined as the 1/e value?

The value 1/e of the spatially exponential PL decay is a universal length scale that has nothing to

do with any diffusion model (linear or nonlinear). This is a standardly used method to study diffusion length in which an exponential fitting is used to extract information about the exciton propagation (Choi *et al.*, Science Advances 6, no. 39 (2020)).

The diffusion equation is a well-defined differential equation. The diffusion length is defined as the Gaussian width in the linear region.

The reviewer insists on this point but we never claimed to have used a diffusion equation. Instead, the reported diffusion length is an experimentally measured quantity that we defined as the $1/e$ value of the spatially exponential decay of PL. It seems the reviewer objects that we named this spatial decay rate of the exponential fit as the “diffusion length”, which would make this concern only about the semantics. Would the reviewer prefer an alternative nomenclature for this experimentally measured quantity?

We agree with the reviewer that using a linear diffusion equation to analyze our experimental data would be incorrect, that’s why we refrained from doing it.

(6) The authors also mention that “We agree with the referee that we are not in the linear regime and we do not assume a linear diffusion equation that is usually employed to extract the diffusion constant”.

The authors studied the diffusion under various excitation powers, with magnitude increased by more than 30 times. The reviewer does not state this has to be a linear or nonlinear region. With this large difference, the authors need to examine it experimentally, and the diffusion length needs to be analyzed accordingly. There may be a linear to nonlinear crossover.

As we mentioned in the previous response, the fact that the magnitude of excitation power changes more than 30 times is precisely the reason why we limited ourselves to experimentally measuring the diffusion length as a function of excitation power and electronic filling factor. We would like to strongly reiterate that **we have not used any rate equation (linear or nonlinear) for extracting parameters related to the diffusion measurements.**

(7) TRPL data in Fig. S3 clearly show two decay times. Why is only the slow component assigned to the exciton lifetime? The fast component is different.

Figure R3. (a) Time-resolved PL intensity showing a bi-exponential fit (black dashed line) to extract the short (τ_S) and long (τ_L) lifetimes at 0V and a peak pump intensity of $45.84 mW/\mu m^2$. (b) Integrated counts with short and long lifetimes as a function of pump power. The counts with a short lifetime are two orders of magnitude lower than the long lifetime counts.

The reviewer correctly points out that there are two decay times. However, from Fig. R3 one can observe that the total counts associated with the short decay time are two orders of magnitude lower than the ones from the long decay time, which makes them negligible for any data analysis. The existence of two decay times has been reported in both multilayer and monolayer TMD systems (Light Sci Appl 10, 72 (2021)) but they are still objects of study.

(8) For the extracted exciton density, the authors obtained 0.5 for excitation power density $56.6 \mu W/cm^2$ and 0.63 for $1888.3 \mu W/cm^2$. If the lifetime is the same as the authors claimed, what does it mean?

It means that excitons are saturable emitters. We show this behavior in Fig. S5 of the supplementary material, where the PL power or the efficiency of exciton creation does not scale linearly with the pump intensity. Instead, it tends to saturate at higher pump powers. Saturable absorption in TMD heterostructures has previously been reported in Chen *et al.*, "Transition-metal dichalcogenides heterostructure saturable absorbers for ultrafast photonics," Opt. Lett. 42, 4279-4282 (2017). This explains the nonlinear scaling of the exciton density with pump power.

Following a suggestion of Reviewer #3 made after the completion of this review round, we performed additional experiments aiming to measure the ratio between intralayer exciton and trion emission intensities. We observe that the transfer time of the electron from the WSe_2 to the WS_2 is so short that it is impossible to detect any emission from intralayer excitons in the bilayer region; see Fig. R4 that displays a wide PL spectrum (including the WSe_2 intralayer exciton/trion energy). At low power (black line), we observe only interlayer exciton X1. Upon increasing power by 3

orders of magnitude, we observe interlayer excitons X1 and X2. However, we do not observe any emission from intralayer exciton/trion.

We consider that this is an important observation that evidences the efficient creation of interlayer excitons in the system, for which we include this information in Sec. VII of the revised Supplementary Material.

Figure R4. Photoluminescence spectrum under different excitation powers in Device D2. The process of creation of interlayer excitons is so efficient, that no PL is detected from intralayer excitons or trions.

Another evidence of the absence of photo-doping in the system can be observed in Fig. R2, where we compare the X1 and X2 emission intensities as a function of power in two different regimes: (1) In off-resonant excitation, the photo-doping effect leads to the reduction of X1 after a power threshold, and (2) in resonant excitation scheme, X1 monotonically increases with power. One can observe the clear difference between these two cases; which allows us to discard any photo-doping effect in our device.